# All-water supercapacitor enabled by 1-nm clay channels

Vasily Artemov [1,2] ✉, Svetlana Babiy [2], Yunfei Teng [2], Jiaming Ma[2], Alexander Ryzhov[3], Tzu-Heng Chen [2], Lucie Navratilova[2], Victor Boureau [2], Pascal Schouwink[2], Mariia Liseanskaia[1], Patrick Huber [1,4], Fikile Brushett [5], Lyesse Laloui[2], Giulia Tagliabue [2] & Aleksandra Radenovic [2]

Water confined to channels one nanometer thick exhibits electrochemical behavior distinct from bulk water, including enhanced protonic conductivity and large dielectric anisotropy. Here, we exploit these characteristics to design a scalable electrochemical energy storage system-a "blue capacitor"-constructed entirely from naturally abundant materials. By assembling layered clays and conductive graphene, we produce 1-nm-thick channels in which confined water acts as the sole electrolyte. We systematically study different clay types, the electrode composition, and separator thickness using complementary physicochemical and electrochemical techniques. The device operates stably up to $1.6 \pm 0.1$ V, achieves specific capacitances of 40 F g$^{-1}$, $97 \pm 2\%$ coulombic efficiency, and stable performance over more than 60,000 charge-discharge cycles at a voltage window of 1 V and a scan rate of 10 mA. Structural and dynamic analyses validate the device architecture, water purity, and proton transport in the nanopores. These results demonstrate that nanoconfined water can function as an electrolyte in a macroscopic electrochemical device, providing a platform for exploring sustainable aqueous energy storage systems.

Water's ability to store and transport electric charge underlies a wide range of processes across technology and nature, from electrochemical devices[1] to lightning in the atmosphere[2] and proton exchange in biological systems[3]. Clouds, for example, accumulate gigajoules of electricity relying largely on water interfaces[4], illustrating that aqueous interfaces can sustain large-scale charge separation (Fig. 1a). Translating this phenomenon into controllable technologies could provide new strategies for sustainable electrochemical energy storage. Yet, despite continued efforts since the pioneering water electrification experiments of Thomson[5] and Tesla[6], artificial systems exploiting pure water as the active electrolyte for reversible charge storage have remained limited. Current batteries and supercapacitors (Fig. 1b) rely on concentrated electrolytes or metal oxides[7,8], which can

limit sustainability and scalability. Developing energy-storage concepts based on abundant and environmentally benign materials is therefore an important objective for next-generation electrochemical technologies[9]. Similar principles may also be relevant for emerging technologies such as biointerfaces and neuromorphic devices[10,11].

At nanometer scales, water's molecular structure and dynamics can deviate from its bulk behavior[12–15]. Under strong nanoconfinement, its dielectric response, proton transport, and interfacial polarization may be significantly modified[16–19]. These effects have become central topics in nanofluidics and interfacial electrochemistry[20–24]. In particular, experiments in artificial nanochannels and layered materials have reported fast proton conduction[18,19] and dielectric anisotropy[16,17], suggesting that nanoconfined water could support efficient charge

[1]Hamburg University of Technology, Hamburg, Germany. [2]École Polytechnique Fédérale de Lausanne, Lausanne, Switzerland. [3]Austrian Institute of Technology, Vienna, Austria. [4]Deutsches Elektronen-Synchrotron DESY, Hamburg, Germany. [5]Massachusetts Institute of Technology, Cambridge, MA, USA. ✉e-mail: vasily.artemov@tuhh.de

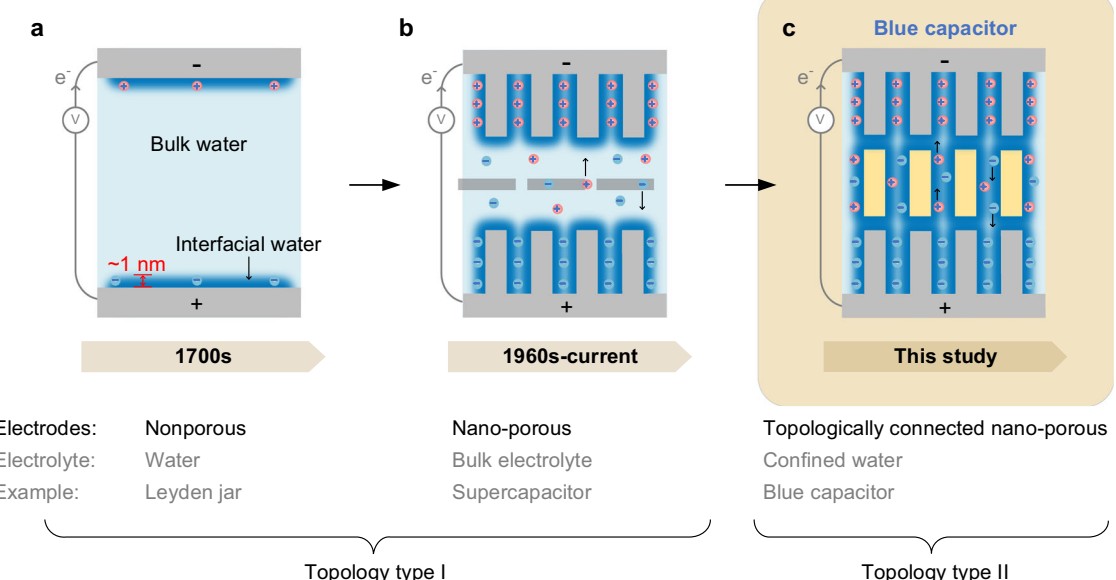

**Fig. 1 | Evolution of double-layer capacitors (DLCs). a** Leyden jar: an early DLC based on water and nonporous electrodes, storing charge in a nanometer-wide interfacial water layer. **b** Supercapacitor: a state-of-the-art DLC with high-surface-area electrodes and a separator immersed in a bulk-like concentrated electrolyte or ionic liquid (light blue), enabling high capacitance at the cost of chemical complexity. **c** Blue capacitor of this study: a new configuration, topologically distinct from the previous two, with a continuous network of strongly confined water as a sole electrolyte (dark blue), continuously crosslinking the electrodes and the separator.

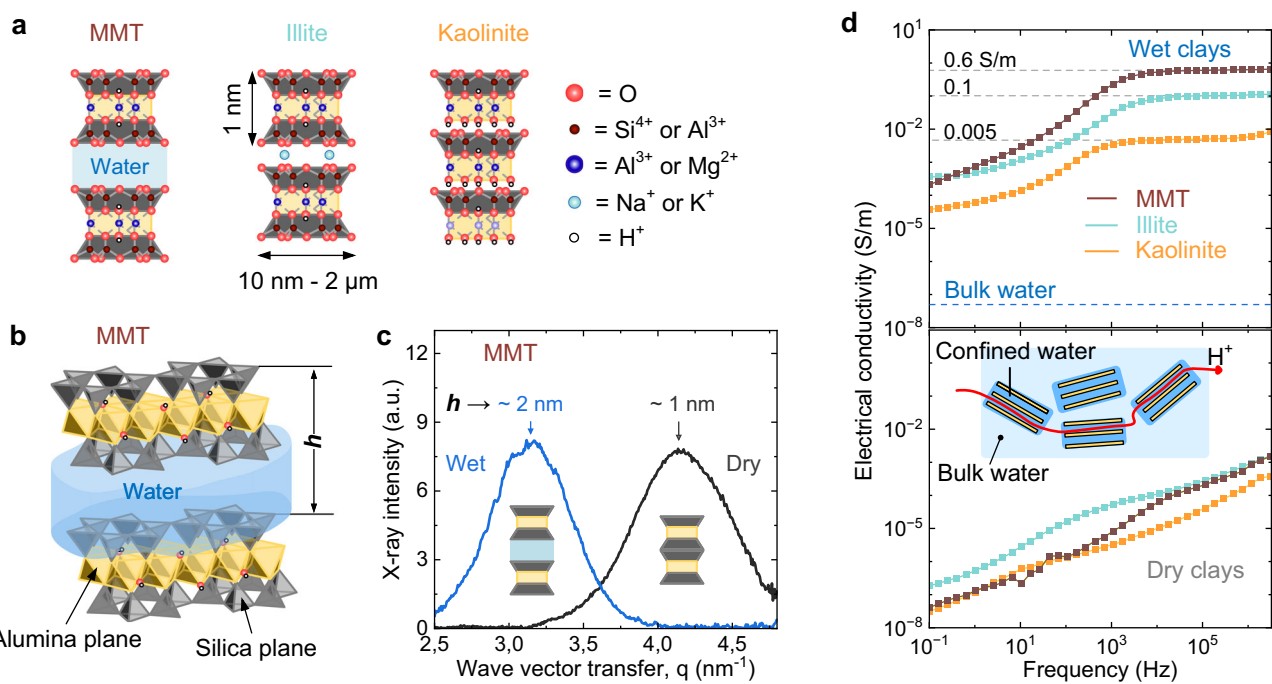

**Fig. 2 | Structure and dielectric properties of clays. a** Crystal structure of the three most abundant natural clays. **b** Enlarged structure of montmorillonite (MMT) clay with an interlayer space accessible to water. **c** Synchrotron small-angle X-ray scattering (SAXS) data for wet and dry MMT (raw data see in SI Fig. S6), showing interlayer water penetration. **d** Proton conductivity of clays under wet (top) and dry (bottom) conditions at $25 \pm 1\,°C$ (see also SI Figs. S12-15). The inset schematically shows the path of a proton through the confined water in wet clays. Source data are provided as a Source Data file.

transport even in the absence of conventional electrolytes (Fig. 1c). However, most studies so far have been limited to nanoscale experimental systems[25,26], which, although valuable for fundamental investigations, face challenges related to reproducibility, scalability, and device integration[26,27].

Natural clays present a unique opportunity to bridge this gap. Composed of layered silica and alumina sheets, they form abundant van-der-Waals heterostructures with interlayer spacings of about 1 nm when hydrated (Fig. 2a). Synchrotron small-angle X-ray scattering shows that hydrated smectite-type montmorillonite expands to accommodate water layers of ~1 nm (Fig. 2b, c), consistent with previous reports[28,29]. These galleries create extended networks of nanometer-scale aqueous channels, which can support proton transport under hydrated conditions[18,19]. Indeed, conductivity

measurements on carefully cleaned clays (Fig. 2d; see "Methods") reveal proton transport comparable to that reported for advanced proton-conducting membranes[30]. Such behaviour is consistent with natural charge-compensation mechanisms in clays[31,32], where mobile ionic species mediate conduction within hydrated interlayer spaces.

Here we combine abundant materials and confined-water electrochemistry to design a scalable electrochemical system-a "blue capacitor"-that uses ultraconfined water as the sole electrolyte. The device integrates clay and graphene into a continuous network of one-nanometer channels, eliminating bulk liquid electrolyte and combining electrodes and separator into a single architecture. The resulting system exhibits robust electrical double-layer capacitance, near-unity coulombic efficiency, and long-term cycling stability exceeding 60,000 cycles without detectable degradation. By exploiting geometric confinement rather than chemical complexity, this design demonstrates that nanoconfined water can serve as a functional electrolyte in macroscopic electrochemical devices.

## Results

### Blue capacitor design

To design the blue capacitor's membrane-electrode unit (MEU), we optimized the nanopore vacuum filtration technique (Fig. 3a; see "Methods"). We controllably self-assembled layer by layer the graphene-clay composite electrodes divided by a pure clay separator, sequentially changing the colloid for the filtration from graphene-clay to clay and to the graphene-clay again (Fig. 3b). We varied the electrode composition and the separator thickness to examine their role in device performance. The result was a scalable, free-standing membrane-electrode unit (MEU) in the form of a nanocomposite film with the thickness ranging from 100 to 200 μm (Fig. 3c), and featuring a three-layer 3D structure (Fig. 3d) with parallel hydrophilic channels of 1 nm (Fig. 3e) without discontinuities at the electrode-separator boundaries. This design mimics nature by focusing on space structuring rather than chemical diversity, unlike previous water-based electrochemical systems with geometrically disjointed electrodes (Fig. 1a, b).

The blue capacitor shows mechanical stability and flexibility (Fig. 4a), typical for vdW heterostructures[33]. Scanning electron microscopy (SEM) energy-dispersive X-ray spectroscopy (EDX) mapping (Fig. 4, b-d) shows the high purity of the cleaned raw materials constituting the device (see "Methods" and SI). Particularly, the presence of only core clay atoms (O, Al, and Si) and graphene (C) has been detected. Electron microscopy close-ups (Fig. 4, e-g) showed that within the layers, material flakes are aligned in highly parallel channels, which intersected at the edges of the domains (Fig. 3i), allowing for the liquid percolation. Domains themselves are aligned within ± 15 degrees. No voids were detected at the interface between the domains across the separator and the electrodes.

### Efficiency of electricity storage in 1 nm water channels

The dry MEU assembly nanochannels were filled with water via capillary condensation from saturated water vapor, excluding contamination, until the weight is stabilized (see "Methods"). To test the blue capacitor's electrical properties, two graphite current collectors were attached to both sides of the MEU (Fig. 5a). The device underwent standard electrochemical tests within a 0–2.1 V voltage window to evaluate charge-discharge cycles (Fig. 5b), current-voltage characteristics (Fig. 5c), and estimate capacitance, longevity (Fig. 5d), Coulombic and energy efficiency (Fig. 5e).

The device showed stable electrochemical performance with no visible degradation after 60,000 cycles at a voltage window of 1 V and a scan rate of 10 mA (Fig. 5e). The stable performance over time, which is usual for supercapacitors, is associated with a lack of chemical reactions, including side reactions, and extreme materials purity, excluding electrode corrosion. Such long cycle stability is characteristic of capacitive systems in which no significant Faradaic reactions occur[34,35]. Variation of electrode composition allowed to estimate an optimal concentration of graphene in the electrodes that was found to be around 35%, as evident from the maximum of the specific capacitance at high electronic conductivity (Fig. 5f). Diminishing the separator thickness is viable to minimize the blue capacitor's Ohmic losses, but is

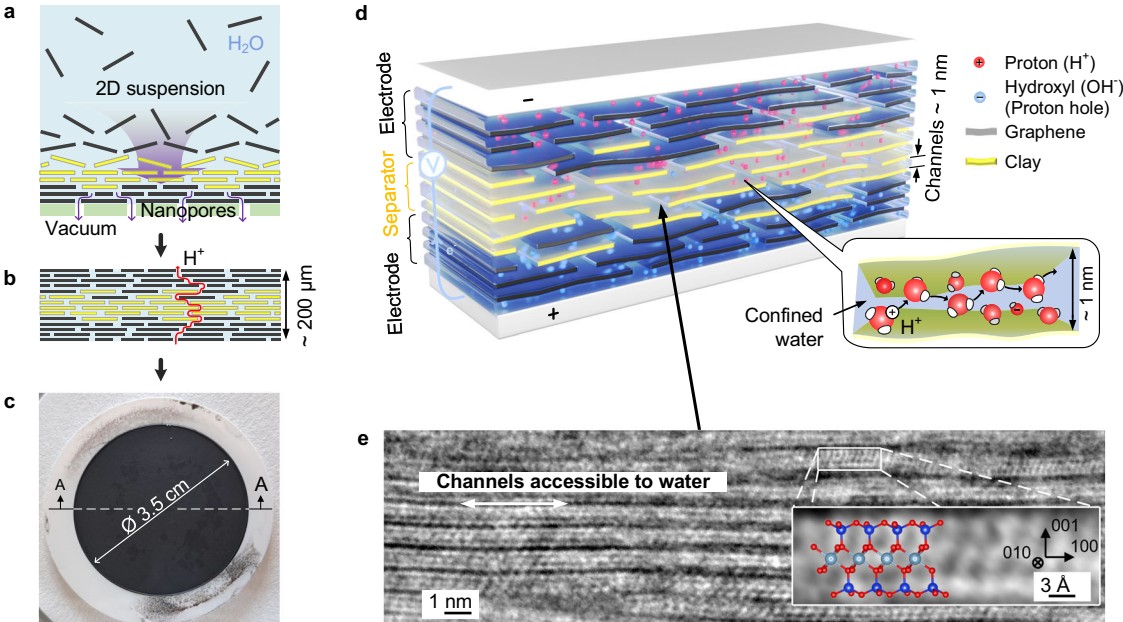

**Fig. 3 | Fabrication of the blue capacitor. a** Vacuum filtration-based assembly of membrane-electrode units (MEUs) from colloidal suspensions. **b** Schematic of the resulting vdW heterostructure composed of graphene-clay composite electrodes and a pure clay separator, forming aligned 1-nm water channels. **c** Photograph of a flexible MEU. **d** Conceptual illustration of the membrane-electrode unit (MEU), built from aligned graphene and clay nanosheets with interlayer water acting as

the sole electrolyte. The inset illustrates the Grotthuss-type proton hopping along a confined water channel, facilitating charge separation across the device. **e** Cross-section imaged by integrated differential phase contrast (iDPC) STEM of the clay separator. The inset shows atomic-resolution imaging and the underlying lattice structure, with oxygen (red), silicon (blue), and aluminum (cyan) atoms.

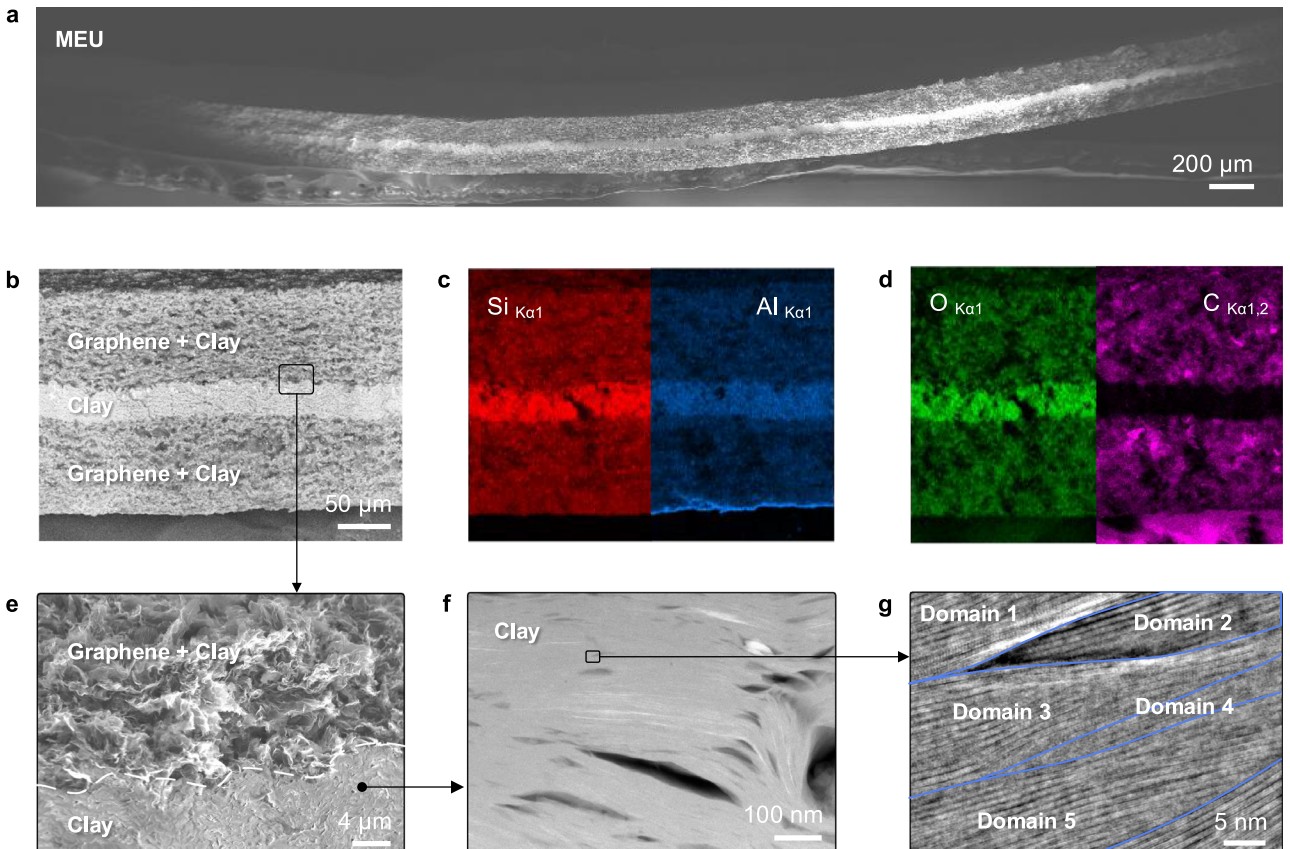

**Fig. 4 | Morphology and chemical composition of the blue capacitor. a** SEM image of a cross-section of the blue capacitor membrane-electrode unit (MEU). **b** Close-up of the SEM image showing a layered structure. **c, d** EDX elemental mapping for Si (red), Al (blue), O (green), and C (magenta), confirming material purity. **e** Further close-up of the SEM image near the electrode-separator contact showing no gap between them. **f** Annular dark-field (ADF) STEM image of a cross-section of the separator showing layered morphology. **g** iDPC-STEM close-up of nanosheet intercalation; blue lines are plotted contours of domains.

limited by short-cutting of the electrodes and requires additional research.

Additionally, we found that confined water has an electrolysis threshold of $1.6 \pm 0.1$ V, evident from the efficiency instability observed at high voltages. The device operates stably up to this threshold before noticeable efficiency losses appear (Fig. 5e). This threshold is higher than the standard 1.23 V one for bulk water at neutral conditions. This yields a capacitance of up to 40 F $g^{-1}$ and a specific energy of around 10 Wh $kg^{-1}$ of electrode material, comparable to that of commercial supercapacitors[36].

## Discussion

### Proton-mediated charge storage in confined water

The electrochemical behaviour of the membrane-electrode units (MEUs) arises from the properties of water confined within the nanometre-scale channels formed between stacked clay platelets and graphene-based electrodes (Fig. 6a). Hydrated clays such as montmorillonite naturally form slit-like galleries with characteristic widths on the order of ~ 1 nm. Under these conditions, water is restricted to thin interfacial layers whose structural and transport properties differ from those of bulk liquid water.

Electrochemical measurements indicate that charge storage in the present system is consistent with electrical double layer (EDL) formation at the graphene-water interface. Several independent observations support this interpretation. First, electrochemical impedance spectroscopy shows the characteristic low-frequency response of a capacitive system, without signatures of Faradaic processes (SI Fig. S33). Second, the introduction of a pH-neutral buffer strongly suppresses capacitance, demonstrating that proton activity is central to the charge-storage mechanism (SI Fig. S32). Third, temperature-dependent impedance measurements yield an activation energy of $0.17 \pm 0.02$ eV (SI Fig. S34), consistent with proton-mediated charge transport in water[37], although this value alone does not distinguish between structural and vehicular diffusion mechanisms. Taken together, the observations suggest that mobile protonic species ($H_3O^+$ and $OH^-$[38]) are likely major charge carriers within the hydrated nanochannels.

Within the clay galleries, water layers are confined to thicknesses comparable to molecular correlation lengths and electrostatic screening distance (Bjerrum length). Such confinement can modify water dynamics, structure, and dielectric properties relative to those of bulk water, thereby enhancing proton mobility along interfacial pathways[18,19]. When an external electric field is applied, protonic species migrate through the hydrated channels and accumulate near the electrode interfaces, where charge is stored via EDL formation (Fig. 6a, panels ii-iii). The transport process is consistent with a relay-race hopping mechanism mediated by confined water molecules, although the present measurements do not directly resolve the microscopic transport pathway.

The critical role of confined water is further demonstrated by control experiments comparing hydrated and dehydrated devices (Fig. 6b). When the clay membranes are dried, the capacitance decreases by several orders of magnitude. Rehydration restores the capacitive response, indicating that water within the nanochannels is essential for both ionic transport and charge storage (Fig. 6c). These results show that the electrochemical functionality of the MEU emerges from the presence of confined water layers rather than from the dry clay framework itself.

While layered clays possess intrinsic structural charge arising from isomorphic substitutions[39], the present experiments indicate that

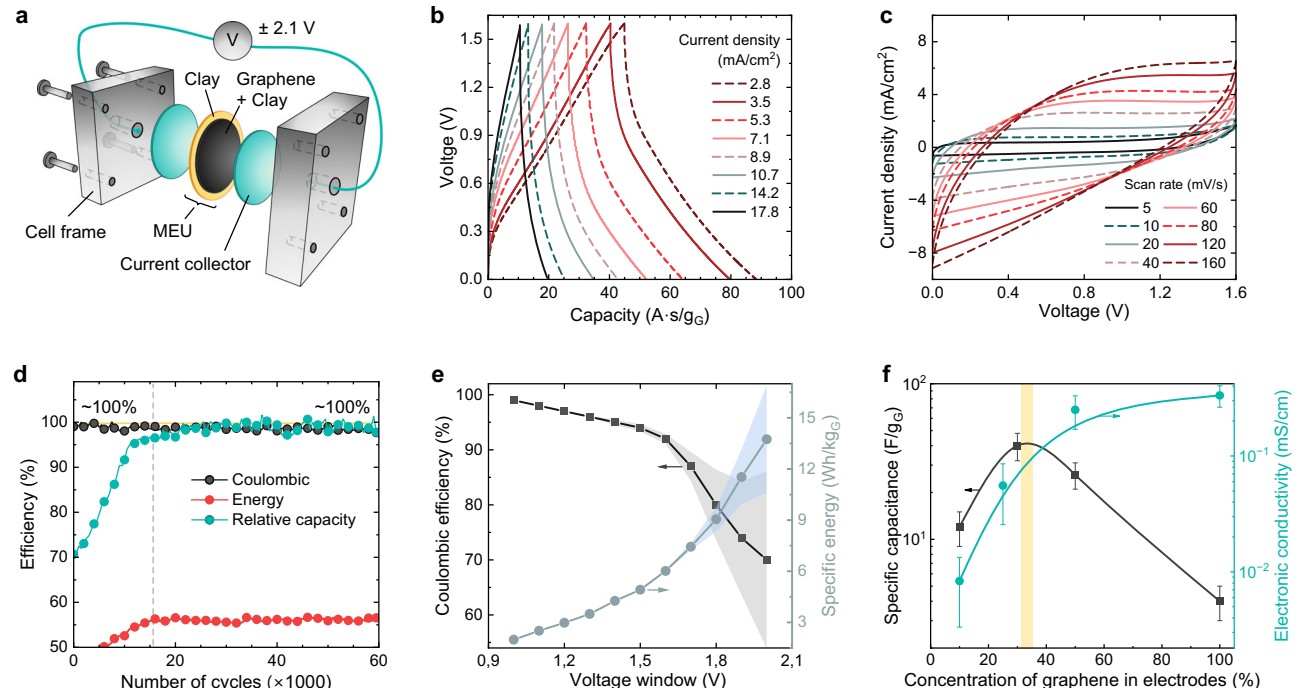

**Fig. 5 | Blue capacitor characteristics. a** Schematic of the MEU measurements assembly. **b** Typical charge-discharge plots at various current densities. **c** Typical cycling voltammograms at various scan rates. **d** Capacity retention, coulomb, and energy efficiency at long-term cycling at 1.6 V and 10 mA. **e** Coulombic efficiency and the energy density vs. voltage window at 8 mA. The error bars show measurement variance at high voltages. **f** Electronic conductivity of electrodes (cyan) and MEU capacitance (black) as functions of graphene concentration in the electrodes. Mean ± SD ($n = 3$ technical replicates). The yellow line shows an optimal region. All data are at 25 ± 1 °C. Source data are provided as a Source Data file.

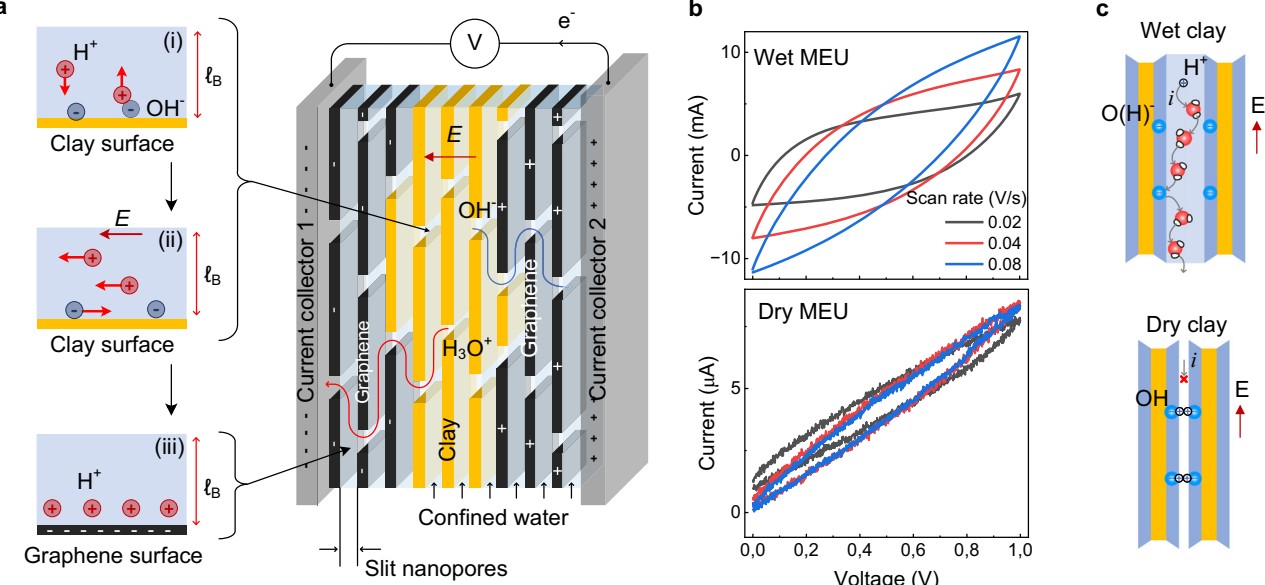

**Fig. 6 | Mechanism of charge storage in hydrated clay nanochannels.**
**a** Simplified schematic of the charge storage mechanism stages: (i) surface-stimulated proton activity; (ii) separation of excess protons and proton holes in an external electric field; and (iii) charge storage via electrical double layer (EDL) capacitance at the graphene-water interface. **b** Cycling voltammograms for wet (top) and dry (bottom) membrane-electrode units (MEUs) at different scan rates at 25 ± 1 °C. **c** Charge separation schematic in case of wet and dry clay channels. Source data are provided as a Source Data file.

structural charge alone does not determine the electrochemical response. Devices assembled from different clay minerals with comparable surface area but varying lattice charge densities exhibit similar qualitative behaviour. This observation suggests that the presence of hydrated nanometre-scale channels is the primary requirement for charge transport, while the clay framework mainly provides a stable host geometry for confined water.

Enhanced protonic transport and modified electrical behaviour of water have been reported in several nanoconfined environments, including artificial nanochannels and layered materials[16,18,19]. While the microscopic origin of these effects may depend on surface chemistry

and channel geometry[40], these studies collectively indicate that restricting water to nanometre-scale environments can substantially modify its electrical response. In the present work, this behaviour is harnessed to realise a macroscopic device in which confined water acts as the sole electrolyte.

## Device implications and sustainable energy storage

Beyond the mechanistic insights, the results demonstrate a device architecture in which naturally occurring layered materials can be used to host a confined water electrolyte. The membrane-electrode unit combines hydrated clay separators with graphene-based electrodes to form a simple layered structure that supports efficient EDL charge storage. The measured specific capacitance, high coulombic efficiency, and long-term cycling stability indicate that this architecture can sustain repeated charge-discharge operation without detectable degradation.

The system differs conceptually from conventional supercapacitors in which ionic transport occurs through bulk liquid electrolytes or organic solvents. In contrast, the present design relies on the transport properties of water confined within nanoscale channels formed by the clay structure itself. As a result, the electrolyte is intrinsically integrated into the separator material and does not require the addition of concentrated salts or organic solvents.

An additional advantage of this architecture lies in the choice of materials. Clay minerals such as montmorillonite are among the most abundant layered materials in the Earth's crust and can be processed at low cost. Combined with carbon-based electrodes and water as the electrolyte, the resulting device consists entirely of environmentally benign and widely available components. While the present work represents a proof-of-concept demonstration, these features suggest potential pathways toward scalable and sustainable electrochemical energy storage systems.

Overall, the present work demonstrates that nanoconfined water hosted in naturally abundant clay structures can function as an effective electrolyte in a macroscopic electrochemical device. By coupling nanoscale interfacial phenomena with scalable materials, this approach provides a platform for exploring sustainable electrochemical systems based on confined aqueous electrolytes. The present results, therefore, establish a phenomenological link between nanoconfined water and electrochemical charge storage at the device level. However, a complete microscopic description of proton transport and interfacial polarization within these channels will require further studies.

In conclusion, we demonstrated a scalable electrochemical energy-storage concept-the blue capacitor-in which ultraconfined water within one-nanometer clay-graphene channels functions as the sole electrolyte. The device integrates electrodes, a separator, and an electrolyte into a continuous network of hydrated nanochannels formed by naturally abundant layered materials. This architecture enables stable proton-mediated charge transport and electrical double-layer capacitance in a system composed entirely of environmentally benign components. The resulting devices exhibit long cycle stability and near-unity coulombic efficiency without the addition of salts or organic electrolytes. These findings show that nanoconfined water hosted in layered materials can support macroscopic electrochemical functionality. Beyond the present proof-of-concept device, this approach may provide a platform for exploring sustainable electrochemical systems based on confined water and aqueous electrolytes.

## Methods

### Sample preparation and cleaning

Clay types, including bentonite, montmorillonite, illite, kaolinite, and talc, were obtained in powder or granular form from Merck and Nagra. Before use, all clays were cleaned using a multistep chemical-free protocol (see SI for details). The use of carefully purified clay suspensions and vapor-phase hydration minimizes dissolved salt content, supporting the interpretation that the observed electrochemical response arises primarily from water confined within the clay galleries. The powders were first milled to submicron sizes, then processed through a combination of repeated dissolution, sedimentation, washing, centrifugation, dialysis, freeze-drying, and sonication cycles. This yielded stable colloidal suspensions of clay nanoparticles in Milli-Q water (resistivity: 18 M$\Omega$ cm), with a concentration of around 0.1 mg mL$^{-1}$, pH=7, and negligible conductivity from contaminants. Electrochemically exfoliated graphene in aqueous suspension was procured from XFNano and used without modification. Nanoporous polymer filters (20–100 nm pores) were sourced from Merck. To ensure chemical purity, pH and conductivity of the water were monitored at each step of the device fabrication and after the measurements (see SI for details). Water was introduced to the clay nanochannels exclusively via adsorption from saturated water vapor, excluding contamination.

### Vacuum filtration

Membrane-electrode units (MEUs) were fabricated via stepwise vacuum filtration of clay and graphene-based suspensions through nanoporous filter substrates. Before filtration, all suspensions were sonicated (1200 W, 20–25 kHz) and equilibrated to 25 ±1 °C. For electrodes, either pure graphene or clay-graphene composite suspensions were used (see SI Table S2). The separator layer was formed from pure clay suspension. The three-layer MEU structure was assembled by sequentially filtering electrode, separator, and counterelectrode suspensions for a total of 5–20 h, depending on the MEU thickness. A water vacuum (40 kPa) pump ensured complete filtration of each layer before proceeding to the next. Filtrate pH and conductivity were monitored to verify the absence of leached species. Unlike dry pressing, vacuum filtration produced dense, layered structures with unidirectionally aligned 2D-like nanosheets held together by van der Waals forces. The resulting membranes were mechanically robust, flexible, and reproducible.

### Electrochemical measurements

MEUs were equilibrated in a saturated water vapor environment for 48 h before testing. Water uptake was confirmed by gravimetric and swelling analysis. Electrochemical measurements, including electrochemical impedance spectroscopy (EIS), cyclic voltammetry (CV), galvanostatic charge-discharge (CD), and long-term cycling, were conducted using BioLogic SP-300 and CH Instruments potentiostats. Preliminary measurements explored a 0–2.1 V range; subsequent long-term tests employed a 0–1.6 V window, chosen based on hydrogen evolution onset observed via CV and CD (SI Figs. S26,27). Graphite plates of 3 cm in diameter and 2 mm thick served as current collectors. All MEUs were sealed with Parafilm to prevent evaporation. The sample mass remained unchanged for about one year after fabrication. Contact pressure was applied via a custom-built through-spring holder to ensure a reproducible electrical connection similar from test to test.

### X-ray diffraction measurement

X-ray Diffraction (XRD) analysis was conducted on a Bruker D8 Discover diffractometer (Bragg-Brentano geometry, K$\alpha_1$ radiation, $\lambda$ = 1.5406 Å) and at the P62 SAXS/WAXS beamline of PETRA III, DESY. Both clay powders and vacuum-filtered membranes were tested. For Bruker measurements, dry samples were prepared by vacuum annealing (24 h), and wet samples by exposure to saturated water vapor (48 h). At PETRA III, wet samples were directly hydrated. Swelling-induced interlayer expansion upon hydration was confirmed in both lab and synchrotron XRD (SI Fig. S8).

### Scanning electron microscopy and EDX

SEM and energy-dispersive X-ray spectroscopy (EDX) were performed using a Zeiss CrossBeam 550 SEM/FIB microscope equipped with an

Oxford Instruments Ultim Max detector. Imaging was conducted at 3-5 kV (SI Fig. S29); EDX mapping at 15 kV (Fig. 4, c and d; SI Fig. S10). For TEM sample preparation, lamellae were fabricated via focused ion beam milling on a Zeiss NVision 40 using $Ga^+$ ions at 30 and 5 kV (SI Fig. S30).

## High-resolution STEM imaging

Atomic-resolution STEM was performed on an aberration-corrected FEI Titan Themis[3] operated at 300 kV. A convergence semi-angle of 8 mrad and beam current of 10 pA were used, with a dwell time of $2\,\mu s$, yielding an electron dose of 460 $e^-\,Å^{-2}$. Simultaneous annular dark-field (ADF) and integrated differential phase contrast (iDPC) images were collected. ADF images used a 17-104 mrad collection angle; iDPC images used 4-15 mrad. iDPC STEM enabled visualization of the light-element atomic lattice while minimizing beam-induced damage[41].

## Data availability

The source data generated in this study are provided in the Source Data file. Source data are provided with this paper.

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

## Acknowledgements

We thank Keith Stevenson, Nianduo Cai, Artem Pronin, Maria Nieves López Salas, Anton Andreev, and Sergey Churakov for fruitful

discussions, Sylvio Haas (DESY) for help with the synchrotron X-ray scattering experiments, Vytautas Navikas for help with the representation of the figures, Milad Sabzehparvar for assistance in the lab, Glaudia Goy, Alexander Petrov, and Samira Afsharian for help with Raman and optical measurements, Fanni Juranyi for insights into neutron scattering analysis, and Manuel Brinker for help with impedance analysis. We thank the EPFL Center for Electron Microscopy (CIME) for access to electron microscopes and the Center for Molecular Water Science (Hamburg) for the close collaboration on related topics.

## Author contributions

V.A. and S.B. conceptualized the presented idea. V.A. proposed and directed the research with the help of F.B., A.Ra., G.T., and L.L.; S.B. prepared the clay materials and characterized their optical, transport, and electrical properties together with V.A. and under the supervision of L.L.; Y.T. developed the vacuum filtration protocol and assembled the membranes and membrane-electrode units with V.A.; J.M. and V.A. conducted the electrochemical measurement under the guidance of G.T.; A.Ry. carried out the data analysis and additional experiments during the review rounds; T.-H.C. develop the protocol and processed the raw materials, controlled their purity, and measured $\zeta$-potentials; L.N., V.B., and Y.T. conducted electron microscopy characterization; P.S. conducted X-ray diffraction experiments; M.L. and P.H. conducted synchrotron radiation-based X-ray scattering experiments and analyzed the data; V.A. wrote the manuscript; and all authors contributed to discussions.

## Funding

V.A. and P.H. disclose support for the research and publication of this work from Deutsche Forschungsgemeinschaft (DFG, German Research Foundation) as part of the Excellence Strategy of the Federal Government and the federal states - EXC 3120/1 BlueMat: Water-Driven Materials [grant number 533771286]. V.A. discloses support for the research of this work from Joachim Herz Stiftung [grant numbers 950002 and 1MR133 (BlueBattery)]. Y.T. discloses support for the research of this work from the Swiss National Science Foundation through the National Centre of Competence in Research Bio-Inspired Materials [grant number 200021-192037]. A.Ra. and T.H.C. disclose support for the research of this work from the European Research Council [grant number 101020445-2D-LIQUID]. S.B. and L.L. disclose support for the research of this work from the Swiss National Science Foundation [grant number 200021-204099, Division II]. M.L. discloses support for the research of this work from the Deutsche Forschungsgemeinschaft (DFG, German Research Foundation) in the DFG Graduate School GRK 2462 'Processes in natural and technical Particle-Fluid-Systems (PintPFS)' [project number 390794421] and the project 'Aqueous Electrolytes in Nanoporous Media: Structure, Dynamics and Electrochemo-Mechanical Actuation' [grant number 509293944]. G.T. discloses support for the research of this work from the Swiss National Science Foundation through Eccellenza [grant number 194181] and Starting Grant [grant number 211695]. Open Access funding enabled and organized by Projekt DEAL.

## Competing interests

The authors declare no competing interests.
