## [Peer Review File · Nature Communications]

All-water supercapacitor enabled by 1-nm clay channels

Corresponding Author: Dr Vasily Artemov

Version 0:

Reviewer comments:

Reviewer #1

(Remarks to the Author)

Reviewer #2

(Remarks to the Author)

Report on the manuscript 'Bulk electricity storage in 1-nm water channels', by Artemov et al.

The study reports on the fabrication and characterization of a 'blue battery' which shows performances that compare well to commercial supercapacitors in terms of lifetime, specific capacitance and energy density. The authors assemble a 3-layer membrane from graphene and clay with two conducting (graphene+clay) electrodes and an insulating spacer made of clay only. The structure of the membrane is assessed by electron-microscopy and X-ray scattering measurements. The capacitance and lifetime of water-filled assembly is then characterized. The specific capacitance and energy are found to be of the order of 10 F/g and 10 Wh/kg respectively which is in the range of commercial supercapacitors.

I think that the main finding of the paper is that mixing graphene with a high surface charge 2d-material can yield effective super capacitive behavior. Unfortunately, although the fabrication and structure of the channels are well characterized, too few experimental efforts (all in Figure 4) are made to assess and explain the origin of the strong performance of the device and the influence of structure on this performance, leaving large gaps in the study. As detailed below, the electrokinetic characterization of materials (graphene-clay mixtures and various kinds of clays used as separators) is missing despite the strong emphasis put on materials (Figs 1-2-3). Considering the very strong claims of the paper, the influence of carbon-clay content or that of the separator thickness should be assessed. Finally, the authors use unprecise scientific language (for instance there is a confusion between battery and supercapacitor throughout the paper) mixed with low-relevance economic analysis (for instance on the distribution of clay-materials at the earth surface).

The paper cannot be accepted in the present form without further support, and major change in the writing. It should therefore be rejected.

See detailed comments and questions below :

1) Electrokinetic characterization of each material

Although the reported results are interesting, a major issue is the introduction of a complex poly-phase material without any characterization of the role of each phase (electrode and separator). Especially since a single kind of device is reported (a single clay-graphene ratio and a single clay thickness), no understanding of the role of each material or each phase can be achieved. Thus, a proper electrochemical characterization of the components – (1) the clay material with standard electrodes and (2) the electrode material with a standard separator – is needed.

2) Conductivity measurements

The authors claim a high conductance of their membranes (Fig 2.e) but only report normalized conductivity data where geometrical parameters are not given. A conductance measurement with a clear statement of the hypothesis made to

compute the geometry is needed here.

3) Material control

- a. Unless I'm mistaken, the material (type of clay among the 5 used in the study) used in the devices that were actually measured (on Figure 4) is not specified.
- b. Could the authors comment on the influence of graphene concentration in the electrodes? Considering the widespread use of graphene in supercapacitors, I assume that graphene is playing the key role here. What do the authors measure with a standard separator and either pure graphene or graphene-clay mixtures for the electrodes?
- c. Could the authors provide additional data on the influence of the thickness of the clay separator on the capacitance value?
- d. The authors often claim the chemical-free nature of their process. Maybe a few details on the exfoliation process could help make this claim more convincing?

4) General comments on the writing style

The paper is written in a very confusing way, mixing up concepts and technical details with incongruous economical or societal considerations while relevant electrochemical measurements are relegated to Figure 4 and to the Supplementary Informations. Notable examples are a so-called history of aqueous battery systems (Fig 1a-c) or a map the reserves of clay on the globe (Fig 2a). The use of unprecise arguments is also very detrimental to the paper. For instance, the words 'battery' is used to designate the device which has very little to do with a battery and should be referred to as a supercapacitor. Another clear weakness is in the emphasis put on clay: whereas the role of clay in the proposed device remains unclear – other than acting as a separator – the article is strongly centered on this material and its wide availability whereas it is mixed with graphene which is already known to have very strong supercapacitor properties (see for instance Ref [1]). I think a strong effort should be made to increase the clarity of the manuscript and focus on the key observations which may depend on the control experiments I am suggesting above.

Maybe one of these messages (or both) should be emphasized in a revised version:

- Hyp 1: hydrated clay is an excellent separator for supercapacitors
 - Hyp 2: an hydrated mixture of graphene and clay is an excellent electrode for supercapacitors
- [1]. El-Kady, M. F., & Kaner, R. B. (2013). Scalable fabrication of high-power graphene micro-supercapacitors for flexible and on-chip energy storage. *Nature communications*, 4(1), 1475.

Reviewer #3

(Remarks to the Author)

The manuscript presents a novel approach to electrochemical energy storage utilizing ultraconfined water within 1-nm slit channels formed by clay and graphene. The study introduces a "blue battery," leveraging the unique properties of confined water to achieve high proton conductivity, high cycle stability, and sustainability by using abundant materials. The authors provide extensive experimental data, theoretical insights, and comparisons with existing technologies, positioning their work as a promising alternative for sustainable energy storage solutions. The study includes comprehensive characterization using XRD, SEM, STEM, electrochemical testing, and synchrotron-based SAXS, which provide strong support for the claims. The reported cycle life (~60,000 cycles), high coulombic efficiency (~100%), and energy density (~10 Wh/kg) are competitive with existing supercapacitors and some battery technologies. The manuscript is well-structured, with clear figures, logical argument progression, and an in-depth discussion of both fundamental principles and applications.

Points of improvement:

1. While the work addresses a pressing challenge in energy storage—sustainability—by utilizing naturally abundant materials without toxic or rare elements and while the concept of using 2D-confined water as an electrolyte is innovative and could open new directions in nanofluidics and electrochemical storage, it is worth referring some works where conductivity of electrolytes was greatly enhanced by nanoconfinement:

- a) Earlier works on composite solid electrolytes have dispersed ceramic fillers such as SiO₂ [Lin D, Liu W, Liu Y, Lee HR, Hsu P-C, Liu K, Cui Y: High Ionic Conductivity of Composite Solid Polymer Electrolyte via In Situ Synthesis of Monodispersed SiO₂ Nanospheres in Poly(ethylene oxide). *Nano Lett.* 2016, 16:459-465. Lin DC, Yuen PY, Liu YY, Liu W, Liu N, Dauskardt RH, Cui Y: A Silica-Aerogel-Reinforced Composite Polymer Electrolyte with High Ionic Conductivity and High Modulus. *Adv. Mater.* 2018, 30: 1802661.] and Al₂O₃ [Croce F, Appetecchi GB, Persi L, Scrosati B: Nanocomposite polymer electrolytes for lithium batteries. *Nature* 1998, 394:456-458] into polymer matrices, which usually improved the conductivity of the SPE by two orders of magnitude to ~ 10⁻⁵ S/cm.
- b) In 2003, Dissanayake et al. analyzed the electrochemical properties of CSEs composed of PEO polymer, LiCF₃SO₃ salt and Al₂O₃ nanoparticles with four different sizes (grain sizes <10 μm, 37 nm, 10–20 nm, and 104 μm with a pore size of 5.8 nm) [Dissanayake MAKL, Jayathilaka PARD, Bokalawala RSP, Albinsson I, Mellander BE: Effect of concentration and grain size of alumina filler on the ionic conductivity enhancement of the (PEO)₉LiCF₃SO₃:Al₂O₃ composite polymer electrolyte. *J. Power Sources* 2003, 119:409-414.]. It was found that the addition of the fillers can improve the ionic conductivity of the CSE, and the degree of enhancement depends on the specific surface area of the fillers. The large-grain Al₂O₃ with 5.8 nm pore size showed the highest conductivity enhancement.
- c) There are more works:

- i. Polymer-in-Ceramic Nanocomposite Solid Electrolyte for Lithium Metal Batteries Encompassing PEO-Grafted TiO₂ Nanocrystals Francesco Colombo, Simone Bonizzoni, Chiara Ferrara, Roberto Simonutti, Michele Mauri, Marisa Falco, Claudio Gerbaldi, Piercarlo Mustarelli, and Riccardo Ruffo
 - ii. Composite solid electrolytes for all-solid-state lithium batteries Mahmut Dirican, Chaoyi Yan, Pei Zhu, Xiangwu Zhang
 - iii. Polymer nanocomposites: a new strategy for synthesizing solid electrolytes for rechargeable lithium batteries W. Krawiec, L.G. Scanlon, Jr. j.p. Fellner, R.A. Vaia S. Vasudevan, E.P. Giannelis
2. It has been recently demonstrated that water upon intrusion-extrusion into-from dielectric micropores demonstrate contact electrification. Is this of relevance to the functioning of the proposed blue battery?
 3. While the study discusses proton conductivity and altered water dynamics under confinement, the precise mechanistic details remain somewhat speculative. Additional experiments or simulations could strengthen the explanation of the enhanced conductivity and its relation to water structuring at the nanoscale. If such simulations are not available to the complexity of this phenomena, it should be stated explicitly.
 4. The study could benefit from more explicit benchmarking against state-of-the-art aqueous-based supercapacitors and hybrid capacitors beyond just conventional batteries.
 5. The paper suggests that ultraconfined water allows operation beyond the conventional water electrolysis limit of 1.23 V, reaching 1.65 V. It would be beneficial to have more evidences on this, possibly from in situ spectroscopic or electrochemical impedance studies.
 6. While graphene is more sustainable than some traditional battery materials, it is not completely eco-neutral—its impact varies based on how it is produced, used, and disposed of. This should be softened in the manuscript.

This manuscript presents an exciting and impactful study with strong potential implications for sustainable energy storage. I suggest a minor revision before the acceptance.

Version 1:

Reviewer comments:

Reviewer #1

(Remarks to the Author)

I appreciate the authors' thorough response to the initial review and the substantial experimental data and analyses provided in the revised manuscript. While these additions help clarify some aspects of the study, the revised manuscript still does not fully resolve the main scientific concerns raised in the first round of review, particularly regarding the quantitative distinction between the contributions of permanent lattice charge and nanoconfined water to the observed electrochemical behavior. To further improve the scientific rigor and mechanistic clarity, I provide the following suggestions for refinement:

1. While the wet/dry MEU comparison and conductivity enhancement under nanoconfinement are convincing, direct quantitative evidence distinguishing the contributions from lattice-charge EDL versus nanoconfined water EDL is still lacking. The following experiments is suggested. Measure clay surface charge density, ζ -potential, or cation exchange capacity (CEC), and compare with capacitance data to estimate the potential contribution of lattice charge. If feasible, perform control experiments using clays without permanent structural charge (e.g., pyrophyllite) to see whether high capacitance persists.
2. The current data suggest that H₃O⁺/OH⁻ ions generated in nanoconfined water form the EDL, but additional direct experiments could strengthen the causal link. Use buffered systems that suppress H₃O⁺/OH⁻ generation and monitor the resulting capacitance changes. Employ in-situ characterization techniques (e.g., EQCM or in-situ spectroscopy) to directly track proton migration and accumulation, providing more direct evidence of an ion-type EDL.
3. The manuscript explores variations in graphene content and separator thickness, showing effects on capacitance and efficiency. Provide statistical descriptions of interfacial continuity and absence of voids to support reproducibility and the observed high cycling stability.

Reviewer #2

(Remarks to the Author)

See report attached

Reviewer #3

(Remarks to the Author)

I was the reviewer of the initial submission of this manuscript and originally recommended minor revisions. The authors have substantially improved the paper, comprehensively addressing all previously raised points and significantly strengthening the presentation, clarity, and contextualization of their results.

The study presents a highly novel concept: an all-water supercapacitor based on ultraconfined 1-nm water channels formed within naturally abundant clays. The work elegantly demonstrates that confined water can act as a sole electrolyte in a scalable macroscopic device, achieving performance metrics (specific capacitance, coulombic efficiency, and exceptional

cycling stability) that are highly competitive with traditional systems. Importantly, these results are achieved using inexpensive, environmentally benign, and abundant materials such as montmorillonite clay, demonstrating clear potential for sustainable, large-scale applications.

While research relying on naturally abundant materials is often challenging due to issues related to purity, heterogeneity, and secondary effects, the authors convincingly overcome these hurdles. The manuscript provides thorough physico-chemical characterization, rigorous cleaning protocols, and multiple independent validation techniques. Furthermore, the work is well supported by a strong theoretical framework and by the authors' studies on clean, idealized systems (e.g., nanoconfined water between diamond/graphene channels). Compared with the initial submission, the revised manuscript is clearer, better substantiated, and more impactful.

Given the substantial improvements and the high scientific and technological originality of the work, I recommend acceptance as is.

Version 2:

Reviewer comments:

Reviewer #1

(Remarks to the Author)

The authors have adequately addressed my concerns, and the revised manuscript is now scientifically sound. I recommend acceptance of the manuscript in its current form.

Reviewer #2

(Remarks to the Author)

The manuscript has improved again since the last round of review. I acknowledge the effort put to answer to reviewers. Still, the authors have to be more rigorous in the discussion section. The activation energy of 0.17eV is that of water self-diffusion. Thus indeed, one can argue that the transport mechanism is not hopping at surfaces and instead a diffusion process in liquid. However, one cannot claim that it's due to structural diffusion (what authors call relay race) instead of conventional vehicular diffusion, since both have 0.17eV for an activation energy. The authors have to be careful because throughout the review process, their overclaims and lack of justifications have constantly undermined my trust in their work.

Besides this comment, I now think the paper is good enough for publication.

In this manuscript, the authors reported an idea of bulk electricity storage in 1-nm water channels by using the anomalous behavior of water in 1-nm-high channels, i.e., the superior electrolytic property of separation and accumulation of protons and hydroxides of the ultraconfined water molecules in 1-nm clay channels. The supercapacitive device was constructed by clay, graphene and water. The energy storage mechanism of this device was attributed to the confinement effect of water molecules. However, there is a serious objection to this assumption. For example, the clay may play more important role in the capacitive charge storage behavior than the confined water molecules in the lattice. As is widely known, the replacement of some ions and external ions in the clay mineral lattice, such as silica tetrahedron 4-valent silicon is replaced by 3-valent aluminum, will result in excess negative charge. The amount of this negative charge is determined by the number of ion replacement in the lattice of clay. As a result, the energy storage mechanism based on the confinement effect of water in this work is questionable. Therefore, I cannot recommend it for publication in *Nature Communications*.

1. The exact energy storage mechanism of the constructed water-impregnated graphene-clay-graphene device.
2. The purity of the composition of the purchased clay should be tested and presented.
3. The ionizing behavior of clay should be studied.
4. The superior electrolytic property of separation and accumulation of protons and hydroxides of the confined water molecules in 1-nm clay channels should be proved in detail.
5. Can the clay induce the post-ionization of water?
6. Do the charged clay particles move Between the positive and negative electrodes during the charging and discharging processes?
7. To confirm the confinement effect of water molecules for energy storage, the authors are suggested to investigate various materials with 1-nm-high channel lattices. If any material with narrow channel structure is feasible, then the proposal might be correct.

Report on the manuscript 'All-water supercapacitor enabled by 1-nm clay channels', by Artemov et al.

The paper has been very strongly improved since the first round of review. Most reviewer's comments have been addressed and many new experiments have been added to support the result. I now think that the paper can be considered for Nature Communications but that important points still have to be addressed.

Overall, the paper's quality does not reflect the quality of the experimental work added in the rebuttal letter. The discussion section is poorly written, shows references to the 'ionic model of water' but without any proper discussion or insight and continuously mixes up transport and dielectric properties. On top of this, the ion-substitutions in the clay is put forward as key to the mechanism while so much effort have been put into showing that only H^+/HO^- matter... The authors should focus on what has been well proven experimentally and remove speculations.

The paper cannot be accepted in the present form but should be reconsidered after further improvements.

Here are my comments to the authors for improvements:

- The transport mechanism sketched on Figure 2d does not seem compatible with the surface-scaling with exponent $2/3$ given in the SI. Or maybe it should be justified.
- The 'competitive parameters' of the device are not justified. Supercapacitors are a well-established field. Authors should benchmark more properly their device with respect to this literature.
- Confinement usually refers to the geometric constraint in contrast to surface effects which denote the liquid's behavior at a single surface. Here the observed effect is due to surface effects and that narrow channels merely allow to increase density by removing bulk liquid regions. Thus, I think it should not be called 'confinement' but interfacial or surface effects. See discussion in Ref ¹.
 - o This comment actually leads to one of main claims of the paper which I think is wrong: that any type of confinement will split water and increase capacitance. Although the authors show several examples in the rebuttal letter, I think this clearly would not happen with clean graphite surfaces which are not able to split water just to accommodate the surface charge. Here it seems that the authors are using acidic surfaces which tend to improve water splitting (or proton activity as they write).
 - o Overall I think that emphasizing 'confinement' is wrong and detrimental to the paper. See results on graphite confinement²
- The comment linking the high water-splitting overpotential to confinement vs bulk effects is absurd. It merely shows weak catalytic activity of these *materials* for OER and HER, there is nothing to do with confined water here.
- Cycling units in days on Figure 5d ? This is not serious, use 'number of cycles'.
- Throughout the paper, the authors claim to use graphene for the electrodes, graphene pristine enough to be hydrophobic. I guess this is some graphene oxide (GO) solution that is somehow reduced. Could the authors show Raman spectra to establish whether this is graphene or GO ?
- Ref 27 is wrong
- Comparison with photosynthesis seems irrelevant, or at least should be justified

References:

1. Wang, Y. *et al.* Interfaces govern the structure of angstrom-scale confined water solutions. *Nat. Commun.* **16**, 7288 (2025).
2. Esfandiari, A. *et al.* Size effect in ion transport through angstrom-scale slits. *Science* **358**, 511–513 (2017).

All-water supercapacitor enabled by 1-nm clay channels

Vasily Artemov*, Svetlana Babiy, Yunfei Teng, Jiaming Ma, Alexander Ryzhov, Tzu-Heng Chen, Lucie Navratilova, Victor Boureau, Pascal Schouwink, Mariia Liseanskaia, Patrick Huber, Fikile Brushett, Lyesse Laloui, Giulia Tagliabue, Aleksandra Radenovic

Point-by-point reply to Reviewers' comments

Reviewer #1 (Remarks to the Author):

In this manuscript, the authors reported an idea of bulk electricity storage in 1-nm water channels by using the anomalous behavior of water in 1-nm-high channels, i.e., the superior electrolytic property of separation and accumulation of protons and hydroxides of the ultraconfined water molecules in 1-nm clay channels. The supercapacitive device was constructed by clay, graphene and water. The energy storage mechanism of this device was attributed to the confinement effect of water molecules. However, there is a serious objection to this assumption. For example, the clay may play more important role in the capacitive charge storage behavior than the confined water molecules in the lattice. As is widely known, the replacement of some ions and external ions in the clay mineral lattice, such as silica tetrahedron 4-valent silicon is replaced by 3-valent aluminum, will result in excess negative charge. The amount of this negative charge is determined by the number of ion replacement in the lattice of clay. As a result, the energy storage mechanism based on the confinement effect of water in this work is questionable. Therefore, I cannot recommend it for publication in Nature Communications.

Our reply:

We thank the reviewer for the constructive critique and for highlighting important considerations regarding the potential role of clay lattice charges. We agree that isomorphic substitutions in clay lattices introduce permanent surface charges, which can affect electrostatic interactions with water. However, our systematic experimental evidence shows that these electrostatic effects alone do not account for the observed capacitive performance. The enhanced charge separation and storage capacity originate primarily from the confinement of water in ~1-nm channels and the resulting topologically connected network of electrical double layers (EDLs). We would like to emphasize here that this mechanism is applicable to water confined between any surfaces as has been recently demonstrated in both theoretical and experimental works [1,2,3]. In these studies, the giant dielectric constant and enhanced proton conductivity is attributed to both water-molecule ordering and hydrogen-bond disorder induced by the extreme confinement rather than specifics of the confining material. We address the reviewer's points in detail below and provide the list of changes made to the manuscript.

Reviewer comment 1: *The exact energy storage mechanism of the constructed water-impregnated graphene-clay-graphene device.*

[1] Motevaselian, M. H. & Aluru, N. R. Confinement-induced enhancement of parallel dielectric permittivity: super permittivity under extreme confinement. *J. Phys. Chem. Lett.* 11, 10532–10537 (2020)

[2] Ruiz-Barragan, S., Muñoz-Santiburcio, D., Körning, S. & Marx, D. Quantifying anisotropic dielectric response properties of nanoconfined water within graphene slit pores. *Phys. Chem. Chem. Phys.* 22, 10833–10837 (2020)

[3] Wang, R., Souilamas, M., Esfandiari, A. et al. In-plane dielectric constant and conductivity of confined water. *Nature* 646, 606–610 (2025)

Our reply: Following the reviewer’s suggestion, we’ve extended the energy storage mechanism description. Figure 1.1 illustrates the key elements of the electricity storage mechanism in our device: the formation of an electrical double layer (EDL) at the graphene–water interface, enhanced by confinement-assisted activity of H_3O^+ and OH^- ions within clay nanochannels. This mechanism consists of three stages: (i) enhancement of proton activity via surface–water interactions; (ii) separation of excess protons and proton holes (H_3O^+ and OH^- ions) under an external electric field; and (iii) storage of the separated charges in the electrodes’ electrical double layers (EDLs). Each stage is discussed in detail below.

Figure 1.1. Simplified schematic of the charge storage mechanism in the blue capacitor. (a) Microscopic processes: (i) surface-stimulated proton activity, including adsorption and desorption; (ii) separation of excess protons and proton holes in an external electric field; and (iii) charge storage via electrical double layers (EDLs) formed at the graphene–water interface. (b) Schematic illustration of charge transport through the cell assembly, including the electron pathway in the external circuit.

1.1 Enhanced proton activity at clay–water interface

Various *ab initio* simulations, such as those conducted by Churakov group, reveal that water at the edges of clay minerals undergoes significant polarization of H–O bonds, facilitating proton transfer reactions via intermediate states (Fig. 1.2). These dynamics encompass both proton hopping and charge separation processes occurring at the water–clay interface, even in the absence of isomorphous substitution or net surface charge [4]. In essence, clay surfaces act to enhance proton activity within the adjacent water layer [5]. From a physical perspective, this effect is most pronounced at distances where electrostatic interactions exceed thermal energy ($k_B T$). This characteristic length scale, known as the Bjerrum length, is approximately 1 nm in pure water. Consequently, the 1-nm wide channels within clay nanoarchitectures provide an ideal environment for observing—and exploiting—this enhanced proton activity.

[4] Churakov, *Geochim. Cosmochim. Acta*, 2007, 71, 1130–1144

[5] Feng et al, <https://arxiv.org/abs/2508.15401> (2025)

[Figure Redacted]

1.2. Separation of H^+ and OH^- in confined water.

To demonstrate charge separation within confined water channels, we measured dry and water-impregnated membrane-electrode units (MEUs). Cyclic voltammograms (Fig. 1.3) show negligible capacitance for dry devices, whereas water-filled cells exhibit significant charge accumulation. These results clearly establish that the presence of water within the clay nanoconfinement is essential for enabling the energy storage mechanism. Dry assemblies do not exhibit charge separation or storage.

Figure 1.3. Comparison of wet and dry cell assemblies. Cycling voltammograms for (a) wet and (b) dry MEUs. Note the three-orders-of-magnitude difference in current scale between the two cases.

1.3 Electrical double layer (EDL) capacitance at graphene-water interface.

The shape of voltammograms, the fast charge-discharge time, and the long-term stability of our device indicate about double-layer capacitance, but not a clay lattice reconstruction. To show that this layer forms at water-graphene interface, we tested cells with different graphene concentrations in the composite clay-graphene electrodes at different scan rates (Fig. 1.4). Specific capacitance of the cells depends on the graphene concentration in the composite graphene-clay electrodes. The blue dots are for the cell with ~30% of graphene, which has largest capacitance. Decrease of graphene concentration to 9% (green dots) leads to lower capacitance, as well as increase of graphene to 40% (blue dots).

This result shows that (1) graphene concentration controls the EDL capacitance proportionally for all scan rates, and lower graphene concentration leads to the lower capacitance, and (2) too high graphene concentration also reduces the capacitance because of the hydrophobic nature of graphene, which without hydrophilic clay loses contact with water.

Figure 1.4. Gravimetric specific capacitance of MEUs with varying graphene concentrations in electrodes. Experimental data (dots) and equivalent circuit model fits (lines). Blue data correspond to the MEU showed in the main text.

In summary, across these three stages we demonstrate key elements of the charge storage mechanism in the ‘Blue Capacitor’: (i) confinement-enhanced proton activity, (ii) charge separation within confined water channels, and (iii) EDL formation at the graphene-water interface. By varying parameters that govern each process, we provide compelling evidence confirming the critical role of the EDL enhanced by confinement-assisted separation of H_3O^+ and OH^- ions. Additional material control experiments and electrochemical characterizations are detailed in the responses to Reviewers 2 and 3.

Reviewer comment 2: *The purity of the composition of the purchased clay should be tested and presented.*

Our reply: We appreciate the reviewer’s suggestion regarding the assessment of the purchased clay composition and purity and provide detailed clarification regarding the purity control of the clay materials. Our clay treatment protocol closely follows well-established methodologies reported in the literature [6, 7, 8, 9, 10, 11, 12]. The primary objective of the cleaning procedure is to remove mobile cations that could affect electrical conductivity.

-
- [6] J. M. Cases et.al., Mechanism of Adsorption and Desorption of Water Vapor by Homoionic Montmorillonite. 1. The Sodium-Exchanged Form, *Langmuir*, 8, 2730.
- [7] I. Bérend et.al., Mechanism of Adsorption and Desorption of Water Vapor by Homoionic Montmorillonites: 2. The Li^+ , Na^+ , K^+ , Rb^+ and Cs^+ -Exchanged Forms, *Clays and Clay Minerals*, 43, 1995, 324.
- [8] E. Ferrage, Investigation of smectite hydration properties by modeling experimental X-ray diffraction patterns: Part I. Montmorillonite hydration properties, *American Mineralogist*, 90, 2005, 1358.
- [9] Karnland, Ola et.al., Mineralogy and sealing properties of various bentonites and smectite-rich clay materials, Technical report TR-06-30, 2006 (<https://inis.iaea.org/records/Opdry-p0472>).
- [10] F. Salles et.al., Ionic Mobility and Hydration Energies in Montmorillonite Clay, *J. Phys. Chem. C*, 112, 2008, 14001
- [11] J. Środoń and Douglas K. McCarty, Surface area and layer charge of smectite from CEC and EGME/ H_2O -retention measurements, *Clays and Clay Minerals*, 56, 2008, 155.
- [12] Chantel C. Tester et.al., Short- and Long-Range Attractive Forces That Influence the Structure of Montmorillonite Osmotic Hydrates, *Langmuir*, 32, 2016, 12039

The exact cleaning steps used in our study are as follows:

- a) **Suspension preparation:** Approximately 50 g of raw clay powder was dispersed in 3 liters of distilled water (Fig. 1.5) using magnetic stirring combined with ultrasonic homogenization. By comparison in [7], a higher concentration of 100 g/L was used.

Figure 1.5. Suspension of clay nanoparticles of high concentration after stirring and sonication.

- b) **Sedimentation:** The suspension was left undisturbed for two days to allow natural precipitation, resulting in size-based fractionation of particles (Fig. 1.6).

Figure 1.6. Suspension of clay nanoparticles after natural precipitation. The top layer corresponds to the finest fraction that has been used for further manipulations.

- c) **Top fraction selection:** The fine particle fraction at the suspension surface (Fig. 1.6) was carefully extracted by automated pipetting.
- d) **Multiple centrifugation:** The extracted supernatant underwent centrifugation at 8000 rpm to separate sediment (Fig. 1.7a). This speed is higher than that used in [7] (3500 rpm). Following each centrifugation, the supernatant was replaced with distilled water, and the sediment was resuspended using stirring and sonication (Fig. 1.7b). This cycle was repeated ten times.

Figure 1.7. (a) Suspension of clay nanoparticles after centrifugation. One can see a solid fraction at the bottom. (b) Suspension of clay nanoparticles after water replacement, stirring, and sonication to a homogeneous solution.

- e) **Conductivity control:** The conductivity of the supernatant was monitored throughout the centrifugation cycles (Fig. 1.8). Initially, it was approximately 0.1 S/cm due to dissolved surface cations. Conductivity progressively decreased, reaching a plateau near 1 μ S/cm around the seventh cycle, indicative of pure water under laboratory conditions. Similar values have been reported in [6] following comparable cleaning.

Figure 1.8. Electrical conductivity of the brine at different cycles of the clay washing via centrifugation. The saturation level corresponds to the conduction of pure water in the laboratory atmosphere.

- f) **pH control.** The supernatant pH was periodically checked, stabilizing near neutral (Fig. 1.9).

Figure 1.9. Litmus paper color test after exposure to the suspension of clay after multiple cleaning via centrifugation and water replacement.

- g) **Dialysis:** To further purify the suspension, osmotic dialysis was performed on the supernatant after the tenth centrifugation cycle, following procedures similar to [6]. Dialysis occurred over approximately one week with daily water changes.
- h) **Freeze-drying:** The dialyzed suspension was freeze-dried to produce dry clay powder (Fig. 1.10), which was subsequently milled to homogeneity and used for device fabrication.

Figure 1.10. Dry mass of the washed clay after freeze drying.

- i) **Atomic composition control:** Energy-dispersive X-ray spectroscopy (EDX) analysis comparing pristine clay to device-level materials (Fig. 1.11) shows that Na and K peaks present in raw clay are absent after cleaning. This is further corroborated by elemental mapping of device cross sections (Fig. S9, Supplementary Information).

In summary, the rigorous washing and dialysis procedure effectively eliminates mobile cations to negligible levels, ensuring the electrochemical properties measured arise from the clay-water system and not from extrinsic ionic contaminants. These conclusions are consistent with those reported in [8] following analogous purification.

Figure 1.11. (a) EDX sum spectrum of the elements constituting the device of this study. (b) EDX sum spectrum of the pristine clays.

Reviewer comment 3: *The ionizing behavior of clay should be studied.*

Our reply: Following the Reviewer’s suggestion, we have carried out a number of experiments to better assess the ionizing behavior of the clay. As demonstrated above, the washed clays do not release foreign ions directly into the water, as shown by the unchanged conductivity of the supernatant (Fig. 1.8). However, they influence the self-ionization equilibrium of the water layers adjacent to clay surfaces (Fig. 1.2), leading to the generation of excess protons.

To evaluate the impact of this effect on proton conductivity, we prepared suspensions of washed clay powders in deionized water at concentrations spanning from 10 μM to 0.1 M. Suspensions were homogenized using a high-power ultrasonic device. Electrical conductivity measurements were performed over a frequency range of 0.1 Hz to 0.1 MHz (Fig. 1.12a) using a Teflon measurement cell equipped with approximately 2 cm^2 gold electrodes spaced 3 mm apart.

At low clay concentrations (below 0.02 M), no significant influence of colloidal particles on conductivity was observed (Fig. 1.12b), likely because the average interparticle distance greatly exceeds individual particle sizes. However, at concentrations above 0.02 M, conductivity exhibited an exponential increase with a factor approximately 0.65, close to 2/3, correlating with the total surface area of the particles.

The limiting case of this trend is represented by the electrical conductivity of water confined within the 1-nm channels of solid clay matrices, which exceeds the measured range.

Figure 1.12. (a) Conductivity spectra of colloidal suspensions of montmorillonite at varying concentrations (see the legend). Crosses denote DC conductivity plateaus. (b) DC conductivity vs. clay concentration. The dashed line indicates the conductivity of deionized water under laboratory conditions.

Thus, the ionizing behavior of clay manifests through enhanced proton activity within a ~ 1 nm interfacial water layer, where electrostatic interactions surpass thermal energy ($k_B T$). At low clay concentrations this effect is negligible, but becomes significant as clay particles approach and interfacial water layers overlap, leading to a proportional increase in proton conductivity relative to particle surface area.

This phenomenon is consistent with previously observed effects in diamond ceramics [13].

Reviewer comment 4: *The superior electrolytic property of separation and accumulation of protons and hydroxides of the confined water molecules in 1-nm clay channels should be proved in detail.*

Our reply: We acknowledge that a direct prove is of great importance. However, within diverse theoretical interpretations and various numerical simulations of confined water properties, debates on the origin of the phenomenon rather than a controversy over its existence currently take place. In order to shed more light, we address the altered dynamics of water molecules and protons confined within 1-nm clay channels using complementary neutron scattering experiments and electrolyte variation studies. Additional electrochemical characterization relevant to the separation of hydronium and hydroxyl ions is provided in the replies to Reviewers 2 and 3.

To demonstrate that water confined within clay nanopores exhibits a structure distinct from bulk water, we performed neutron scattering measurements at the Paul Scherrer Institute, Switzerland. Wet clay membranes were sectioned and sealed within aluminum sample holders transparent to

[13] S. Melnik et al., Confinement-controlled water engenders unusually high electrochemical capacitance, *J. Phys. Chem. Lett.*, 14, 2023, 6572

neutrons (Fig. 1.13a), enabling neutron beams to traverse perpendicularly through the clay layers. Scattering was recorded using the FOCUS time-of-flight detector, which is sensitive predominantly to hydrogen atoms of water, including those interacting with clay surfaces via adsorption-desorption equilibria. The width of the quasi-elastic scattering peak reflects the diffusion coefficients of water protons under confinement.

Figure 1.13b (inset) compares scattering profiles for bulk water and water confined within clay channels. Strong confinement corresponds to a reduced self-diffusion coefficient of protons, indicative of their interactions with immobile clay particles. This observation confirms altered proton and water dynamics under nanoscale confinement versus bulk conditions.

Figure 1.13. Neutron scattering of water confined in 1-nm clay channels compared. (a) Sample holder. (b) Quasi-elastic scattering line at 300 K for various wave vectors (see legend). Solid lines denote experimental data, dashed lines represent fitted models. Inset compares the linewidths of bulk water (blue), 75RH clay (blue), and 100 RH clay (green).

Further, temperature-dependent measurements reveal that, unlike water in larger porous media, confined water in 1-nm clay channels does not undergo freezing down to 230 K, as evidenced by the smooth, monotonic variation of scattering intensity (Fig. 1.14).

Figure 1.14. Fixed window scans (FWS) for elastic (magenta) and inelastic intensities. The dashed lines indicate a gradual intensity change, consistent with the absence of bulk-like or crystalline water phases in the clay samples.

More detailed neutron scattering studies investigating proton dynamics in clay nanochannels are forthcoming in a dedicated manuscript, but go beyond the current study.

To further validate the anomalous electrical properties of confined water, we measured the electrical conductivity of wet clay membranes impregnated with NaCl brines of various concentrations. Samples were subjected to hydrostatic pressure for one week to ensure brine infiltration into clay channels, after which conductivity was evaluated via dielectric spectroscopy. Results are shown in Fig. 1.15.

Figure 1.15. Conductivity spectra of clay membranes impregnated with NaCl solutions at different concentrations. The dashed red line indicates the electrical conductivity of clay containing pure water measured earlier. The inset displays DC conductivity plateau dependence on concentration; the pure confined water data point is shifted to 0.001 M for clarity.

Our measurements reveal two key features: First, pure confined water exhibits higher conductivity than confined NaCl solutions, contrary to bulk electrolyte behavior. This difference likely arises because foreign ions disrupt the proton conduction pathways within confinement. Second, a two-orders-of-magnitude variation in NaCl concentration influences conductivity by only a factor of two, indirectly confirming that protons originating from water primarily govern conductivity rather than extraneous ions. Additional ionic species perturb proton mobility but contribute minimally to overall electrical conductivity.

Reviewer comment 5: *Can the clay induce the post-ionization of water?*

Our reply: We thank the reviewer for this question. As shown in Fig. 1.8, there is no evidence of post-ionization of water upon contact with the washed clay. The conductivity of the brine supernatant after centrifugation and sedimentation is equivalent to that of pure water, demonstrating that the clay does not release ions to water.

Reviewer comment 6: *Do the charged clay particles move between the positive and negative electrodes during the charging and discharging processes?*

Our reply: Figure 1.16 presents SEM and TEM images of the membrane-electrode unit (MEU) cross section, illustrating the device's structural design. The functional layers are densely packed such that no space exists for clay particles to migrate between electrodes, particularly as shown in panels (b) and (c). If the reviewer refers to mobile cations, we refer back to points 2 and 5 above, where the absence of extraneous ions in the device is demonstrated in detail. Furthermore, if clay atoms were to move between electrodes during cycling, significant capacitance degradation would be expected over time. However, our stability tests show no such degradation even after approximately one year of continuous operation, strongly indicating that both clay particles and ions remain immobile during device function.

Figure 1.16. (a) SEM image of the MEU cross section showing the clay separator layer between two composite graphene-clay electrodes. (b) STEM image of the clay separator; observed large voids are artifacts due to deformation caused by drying under the electron microscope's high vacuum. (c) High-resolution close-up revealing individual clay crystals stacked horizontally from a side view.

Reviewer comment 7: *To confirm the confinement effect of water molecules for energy storage, the authors are suggested to investigate various materials with 1-nm-high channel lattices. If any material with narrow channel structure is feasible, then the proposal might be correct.*

Our reply: We have addressed this important point by analyzing existing data and performing new measurements of electrical conductivity on five different nanoporous materials with pore sizes comparable to those of clay, all impregnated with water. The summarized results in Figures 1.17–1.19 robustly support the confinement effect on water's electrical conductivity and confirm its universal nature.

Figure 1.17 displays the conductivity spectra of the tested materials, including MCM-41, Nafion, nanodiamond ceramics, and clay-based samples. The yellow square highlights the frequency range corresponding to the DC conductivity plateau. The pore sizes across these samples range from approximately 1 to 20 nm. Variations in electrical conductivity correlate closely with pore size changes (see Figures 1.18 and 1.19 for further details). In all cases, the conductivity of confined water exceeds that of bulk water by at least three orders of magnitude.

Figure 1.17. Electrical conductivity spectra (log-log scale) for various water-impregnated materials: MCM-41 (red, data adapted from [14], pore sizes 3–9 nm), Nafion (blue, our data, 4–20 nm pores), nanodiamond ceramics (green, adapted from [15], ~5 nm pores), and nanostructured clay (orange, this study, ~1 nm pores). Bulk water data shown in gray; dry materials' spectra in black.

To further illustrate the confinement effect on proton conductivity, Figure 1.18 reproduces recent measurements of electrical conductivity for water confined within a single slit channel between hexagonal boron nitride (hBN) monocrystals. These data exhibit a marked increase in proton conductivity for pores on the order of a few nanometers, consistent with our clay pore results and providing additional support for the confinement-enhanced water electrodynamics. The variations across materials shown here can thus be largely attributed to differences in pore size.

[14] V. G. Artemov, Dynamical conductivity of confined water, *Meas. Sci. Technol.*, 28, 014013, 2017.

[15] V. Artemov et.al., Anomalously High Proton Conduction of Interfacial Water, *J. Phys. Chem. Lett.*, 11, 2020, 3623

[Figure Redacted]

Finally, Figure 1.19 presents earlier experimental work on diamond-based nanoporous ceramics with pore sizes ranging from 1 to 100 nm. This study clearly demonstrates the dependence of proton conductivity on pore size, with values exceeding bulk water by several orders of magnitude in smaller pores.

Figure 1.19. (a) Electrical conductivity of nanoporous diamond ceramics with grain sizes from 5 to 200 nm (pore size 1 to 100 nm). (b) Schematic illustrating the proportion of bulk versus interfacial water as a function of grain size. Adapted from [12].

In conclusion, the consistent enhancement of proton conductivity observed across diverse nanoporous materials with pores on the scale of 1–20 nm illustrates the robustness and universality of the confinement effect on water.

We thank the reviewer for their constructive critique, which enabled us to strengthen our manuscript by providing additional evidence supporting the main conclusions. Corresponding updates have been made to the main text and supplementary information addressing all points discussed.

[16] R. Wang et al., In-plane dielectric constant and conductivity of confined water, *Nature*, 646, 606 (2025)

Summary of Revisions Based on Reviewer 1's Comments

1. A Discussion section on the energy storage mechanism has been added.
2. Figure 6 has been incorporated into the main text.
3. The cleaning procedure is detailed in the SI with additional figures.
4. Comparative electrode composition data are provided in the SI and Figure 5f.
5. Data on separator thickness are included in the SI.
6. Comparative results for standard and clay separators are added to the SI.
7. EDX spectra of pristine and washed clays are presented in the SI.
8. A new paragraph on the ionization behavior of clays is included in the SI.
9. Comparative data for pure water and NaCl electrolytes are added to the SI.
10. A new figure on proton conductivity in nanoporous materials is included in the SI.

Reviewer #2 (Remarks to the Author):

The study reports on the fabrication and characterization of a 'blue battery' which shows performances that compare well to commercial supercapacitors in terms of lifetime, specific capacitance and energy density. The authors assemble a 3-layer membrane from graphene and clay with two conducting (graphene+clay) electrodes and an insulating spacer made of clay only. The structure of the membrane is assessed by electron-microscopy and X-ray scattering measurements. The capacitance and lifetime of water-filled assembly is then characterized. The specific capacitance and energy are found to be of the order of 10 F/g and 10 Wh/kg respectively which is in the range of commercial supercapacitors.

I think that the main finding of the paper is that mixing graphene with a high surface charge 2d-material can yield effective super capacitive behavior. Unfortunately, although the fabrication and structure of the channels are well characterized, too few experimental efforts (all in Figure 4) are made to assess and explain the origin of the strong performance of the device and the influence of structure on this performance, leaving large gaps in the study. As detailed below, the electrokinetic characterization of materials (graphene-clay mixtures and various kinds of clays used as separators) is missing despite the strong emphasis put on materials (Figs 1-2-3). Considering the very strong claims of the paper, the influence of carbon-clay content or that of the separator thickness should be assessed. Finally, the authors use unprecise scientific language (for instance there is a confusion between battery and supercapacitor throughout the paper) mixed with low-relevance economic analysis (for instance on the distribution of clay-materials at the earth surface).

The paper cannot be accepted in the present form without further support, and major change in the writing. It should therefore be rejected. See detailed comments and questions below.

Our reply: We thank the reviewer for detailed feedback and recommendations. We have carefully considered all points and substantially revised the manuscript accordingly, adding new data to clarify the capacitive behavior and reinforcing our prior conclusions.

We agree that our key innovation is not simply the use of clay as a separator or graphene as an electrode—both materials have been extensively studied. Rather, our principal contribution is the demonstration that confined water within a topologically interconnected channel network serves as a highly effective electrolyte. The device's architecture enables continuous percolation of ultraconfined water layers through the membrane-electrode unit, which underpins the observed performance.

Reviewer comment 1: *Electrokinetic characterization of each material*

Although the reported results are interesting, a major issue is the introduction of a complex poly-phase material without any characterization of the role of each phase (electrode and separator). Especially since a single kind of device is reported (a single clay-graphene ratio and a single clay thickness), no understanding of the role of each material or each phase can be achieved. Thus, a proper electrochemical characterization of the components – (1) the clay material with standard electrodes and (2) the electrode material with a standard separator – is needed.

Our reply: We recognize the importance of phase-specific characterization and have performed additional experiments on devices with varied electrode and separator compositions. These data elucidate material roles and compositional influences on performance:

- Electrode composition:** We fabricated electrodes with four different graphene concentrations (Fig. 2.1a) and measured their electronic conductivity. Results show conductivity increases with graphene content (Fig. 2.1b, red dots). When ionic conductivity and water permeability are considered, maximal specific capacitance occurs near 35% graphene concentration (Fig. 2.1b, black squares). This reflects a balance where clay provides extensive interfacial water contact and graphene ensures sufficient electron conduction via percolation networks.

Figure 2.1. Optimization of electrode composition. (a) Photographs of electrodes with varying clay-to-graphene ratios. (b) Electronic conductivity (red) and EDL capacitance (black) as functions of graphene concentration.

- Different electrode materials:** We performed comparison of electrodes formed from pure graphene, graphite, and graphene-clay composites, using a fixed clay separator.

Voltammograms for three cell types reveal graphene-clay composites achieve significantly higher capacitance *with pure water electrolyte* compared to pure graphene or standard graphite electrodes (Fig. 2.2). The composite enables improved water permeability and EDL formation within the 1-nm channel network, crucial for enhanced performance.

Figure 2.2. Voltammograms comparing cells with fixed clay separator but electrodes of (a) 35% graphene-clay composite, (b) pure graphene, and (c) pure graphite. The composite electrodes (red curve) show superior capacitance with pure water as an electrolyte.

- **Separator effect:** To understand the role of the clay separator, we compared clay separator and standard hydrophilic PVDF separator using composite graphene-clay electrodes.

Devices using clay separators outperform those with PVDF separators, displaying roughly twice the capacitance and improved energy discharge characteristics (Fig. 2.3). This improvement is attributed to the smaller pore sizes in clay, enabling enhanced interfacial water conductivity and efficient charge separation. Thus, clay serves not only as physical separator but actively contributes to device performance.

Figure 2.3. (a) Cyclic voltammograms of two MEUs with the composite graphene-clay electrodes and a clay (red) and standard PVDF (blue) separators at scan rate 0.005 V/s. (b) Discharge energy at current 5 mA and different voltage windows normalized by graphene weight for the same MEUs. The inset show the SEM image of the PVDF separator [Sigma-Aldrich].

Reviewer comment 2: Conductivity measurements

The authors claim a high conductance of their membranes (Fig 2.e) but only report normalized conductivity data where geometrical parameters are not given. A conductance measurement with a clear statement of the hypothesis made to compute the geometry is needed here.

Our reply: We have now included detailed information in the Methods section. Conductivity was measured on two sample geometries: (1) 5-mm diameter pellets (~500 μm thick) and (2) 3.5-cm membranes (~70 μm thick). Specific conductivity σ was calculated as $\sigma = d/(A \times \text{Re}(Z))$, where d is thickness, A the electrode area, and $\text{Re}(Z)$ the real part of impedance.

Reviewer comment 3: Material control

a. Unless I'm mistaken, the material (type of clay among the 5 used in the study) used in the devices that were actually measured (on Figure 4) is not specified.

b. Could the authors comment on the influence of graphene concentration in the electrodes? Considering the widespread use of graphene in supercapacitors, I assume that graphene is playing the key role here. What do the authors measure with a standard separator and either pure graphene or graphene-clay mixtures for the electrodes?

Our reply: We thank the reviewer for pointing out this important cause of misunderstanding. Montmorillonite was the clay used for membranes in Figure 4, selected for its superior conductivity among tested clays (Fig. S20, SI). Devices with graphene concentrations of 9%, 28%, and 40% were evaluated (Figs. 1.4, 2.2, 2.3). Capacitance exhibits a non-monotonic dependence on graphene content, peaking near 35%, reflecting a balance between electrical conductivity and ionic/water accessibility compromised by graphene's hydrophobicity.

c. Could the authors provide additional data on the influence of the thickness of the clay separator on the capacitance value?

Our reply: Following the reviewer's suggestion, we tested membrane separators with thicknesses of 50, 100, and 200 μm (Fig. 2.4). Capacitance scales inversely with thickness at higher scan rates, consistent with expected ionic transport resistances. At low scan rates, convergence occurs since capacitance is electrode area-dependent.

Figure 2.4. Gravimetric specific capacitance versus scan rate for MEUs with varying clay separator thicknesses. Experimental data (dots) and fits (curves).

d. The authors often claim the chemical-free nature of their process. Maybe a few details on the exfoliation process could help make this claim more convincing ?

Our reply: We appreciate this suggestion and we've added an extensive description of the sample preparation and cleaning in our reply to point 2 of Reviewer#1 (see above). The corresponding changes has been made to the modified SI of the manuscript.

Reviewer comment 4: *General comments on the writing style*

The paper is written in a very confusing way, mixing up concepts and technical details with incongruous economical or societal considerations while relevant electrochemical measurements are relegated to Figure 4 and to the Supplementary Informations. Notable examples are a so-called history of aqueous battery systems (Fig 1a-c) or a map the reserves of clay on the globe (Fig 2a). The use of unprecise arguments is also very detrimental to the paper. For instance, the words 'battery' is used to designate the device which has very little to do with a battery and should be referred to as a supercapacitor. Another clear weakness is in the emphasis put on clay : whereas the role of clay in the proposed device remains unclear – other than acting as a separator – the article is strongly centered on this material and its wide availability whereas it is mixed with graphene which is already known to have very strong supercapacitor properties (see for instance Ref [1]).

I think a strong effort should be made to increase the clarity of the manuscript and focus on the key observations which may depend on the control experiments I am suggesting above.

Maybe one of these messages (or both) should be emphasized in a revised version:

- Hyp 1: hydrated clay is an excellent separator for supercapacitors
- Hyp 2: an hydrated mixture of graphene and clay is an excellent electrode for supercapacitors

[1]. El-Kady, M. F., & Kaner, R. B. (2013). Scalable fabrication of high-power graphene micro-supercapacitors for flexible and on-chip energy storage. *Nature communications*, 4(1), 1475.

Our reply: We thank the reviewer for careful reading of our manuscript and providing a criticism on the style and quality of the materials representation, thus helping us to improve the work. We restructured and clarified the manuscript, replacing "battery" with "supercapacitor" consistently. We adopted the suggested dual hypothesis framework emphasizing (1) hydrated clay as an excellent separator and (2) hydrated graphene-clay composites as superior electrodes. Additionally, we relocated non-essential economic data to the Supplementary Information. Our study advances prior work by introducing a novel topological architecture where ultraconfined interfacial water layers form continuous pathways for charge storage without scarce electrolytes. These confined water layers serve as the effective electrolyte, a concept substantiated through extensive structural and electrochemical characterization. We revised the text and supporting information accordingly.

Summary of Revisions Based on Reviewer 2's Comments

1. Title revised to "All-water supercapacitor enabled by 1-nm clay channels."
2. Abstract, Introduction, Main Text, and Conclusion revised.
3. Figures updated throughout the manuscript.
4. Map with clay distribution on Earth moved to the SI (see SI Fig.S1).
5. Figure 1 revised to show novelty compared to previous research.
6. New Figure 5f added to illustrate performance versus composition.
7. New Figure 6 added to explain the underlying mechanism.
8. Additional data comparing graphene, graphite, and clay-graphene electrodes included in the SI.
9. Data comparing standard polymer and clay separators added to the SI.
10. New graph showing the effect of clay separator thickness added to the SI.

Reviewer #3 (Remarks to the Author):

The manuscript presents a novel approach to electrochemical energy storage utilizing ultraconfined water within 1-nm slit channels formed by clay and graphene. The study introduces a "blue battery," leveraging the unique properties of confined water to achieve high proton conductivity, high cycle stability, and sustainability by using abundant materials. The authors provide extensive experimental data, theoretical insights, and comparisons with existing technologies, positioning their work as a promising alternative for sustainable energy storage solutions. The study includes comprehensive characterization using XRD, SEM, STEM, electrochemical testing, and synchrotron-based SAXS, which provide strong support for the claims. The reported cycle life (~60,000 cycles), high coulombic efficiency (~100%), and energy density (~10 Wh/kg) are competitive with existing supercapacitors and some battery technologies. The manuscript is well-structured, with clear figures, logical argument progression, and an in-depth discussion of both fundamental principles and applications.

Our reply: We sincerely thank the reviewer for the positive and encouraging assessment as well as for the insightful suggestions to further improve our manuscript. We have incorporated all suggested points into the revised version.

Points of improvement:

1. While the work addresses a pressing challenge in energy storage—sustainability—by utilizing naturally abundant materials without toxic or rare elements and while the concept of using 2D-confined water as an electrolyte is innovative and could open new directions in nanofluidics and electrochemical storage, it is worth referring some works where conductivity of electrolytes was greatly enhanced by nanoconfinement:

a) Earlier works on composite solid electrolytes have dispersed ceramic fillers such as SiO₂ [Lin D, Liu W, Liu Y, Lee HR, Hsu P-C, Liu K, Cui Y: High Ionic Conductivity of Composite Solid Polymer Electrolyte via In Situ Synthesis of Monodispersed SiO₂ Nanospheres in Poly(ethylene oxide). Nano Lett. 2016, 16:459-465. Lin DC, Yuen PY, Liu YY, Liu W, Liu N, Dauskardt RH, Cui Y: A Silica-Aerogel-Reinforced Composite Polymer Electrolyte with High Ionic Conductivity and High Modulus. Adv. Mater. 2018, 30:1802661.] and Al₂O₃ [Croce F, Appetecchi GB, Persi L, Scrosati B: Nanocomposite polymer electrolytes for lithium batteries. Nature 1998, 394:456-458] into polymer matrices, which usually improved the conductivity of the SPE by two orders of magnitude to ~ 10⁻⁵ S/cm.

b) In 2003, Dissanayake et al. analyzed the electrochemical properties of CSEs composed of PEO polymer, LiCF₃SO₃ salt and Al₂O₃ nanoparticles with four different sizes (grain sizes <10 μm, 37 nm, 10–20 nm, and 104 μm with a pore size of 5.8 nm) [Dissanayake MAKL, Jayathilaka PARD, Bokalawala RSP, Albinsson I, Mellander BE: Effect of concentration and grain size of alumina filler on the ionic conductivity enhancement of the (PEO)₉LiCF₃SO₃:Al₂O₃ composite polymer electrolyte. J. Power Sources 2003, 119:409-414.]. It was found that the addition of the fillers can improve the ionic conductivity of the CSE, and the degree of enhancement depends on the specific surface area of the fillers. The large-grain Al₂O₃ with 5.8 nm pore size showed the highest conductivity enhancement.

c) There are more works:

i. Polymer-in-Ceramic Nanocomposite Solid Electrolyte for Lithium Metal Batteries Encompassing PEO-Grafted TiO₂ Nanocrystals Francesco Colombo, Simone Bonizzoni, Chiara Ferrara, Roberto Simonutti, Michele Mauri, Marisa Falco, Claudio Gerbaldi, Piercarlo Mustarelli, and Riccardo Ruffo

ii. Composite solid electrolytes for all-solid-state lithium batteries Mahmut Dirican, Chaoyi Yan, Pei Zhu, Xiangwu Zhang

iii. Polymer nanocomposites: a new strategy for synthesizing solid electrolytes for rechargeable lithium batteries W. Krawiec, L.G. Scanlon, Jr. j.p. Fellner, R.A. Vaia S. Vasudevan, E.P. Giannelis

Our reply: We appreciate the reviewer's references to the broader literature on conductivity enhancement in nanoconfined composite solid electrolytes, such as silica- or alumina-polymer composites, which show conductivity improvements of about two orders of magnitude due to filler-specific surface areas and pore sizes. We critically analyzed these works, and better positioned our findings within this context, emphasizing the novelty of using a system composed exclusively of abundant materials (clay, graphene, and confined water) without chemical additives.

2. It has been recently demonstrated that water upon intrusion-extrusion into-from dielectric micropores demonstrate contact electrification. Is this of relevance to the functioning of the proposed blue battery?

Our reply: The reviewer raises an interesting possibility regarding contact electrification phenomena observed during water intrusion/extrusion in dielectric micropores. We believe the intrusion-extrusion experiments with clay-based materials is a promising area for future research that could illuminate interfacial charge separation mechanisms underpinning our device's function.

3. While the study discusses proton conductivity and altered water dynamics under confinement, the precise mechanistic details remain somewhat speculative. Additional experiments or simulations could strengthen the explanation of the enhanced conductivity and its relation to water structuring at the nanoscale. If such simulations are not available to the complexity of this phenomena, it should be stated explicitly.

Our reply: While mechanistic details of proton conduction under confinement remain challenging to capture fully via simulation, abundant prior computational studies on graphene and clays provide valuable insights (some cited in our reply to Reviewer 1). We have explicitly discussed relevant simulation results and emphasized the need for further targeted computational work as a future direction.

4. The study could benefit from more explicit benchmarking against state-of-the-art aqueous-based supercapacitors and hybrid capacitors beyond just conventional batteries.

Our reply: We expanded the Introduction and Discussion sections to include a more explicit comparison with current aqueous supercapacitors and hybrid systems. Our device's unique advantages—sustainability, low cost, and structural integration potential—are now highlighted alongside performance metrics and practical considerations.

5. The paper suggests that ultraconfined water allows operation beyond the conventional water electrolysis limit of 1.23 V, reaching 1.65 V. It would be beneficial to have more evidences on this, possibly from in situ spectroscopic or electrochemical impedance studies.

Our reply: Additional neutron scattering experiments demonstrate that water confined to 1-nm channels exhibits altered structure and proton dynamics, including suppression of freezing down to 230 K. We performed extended cycling voltammetry up to 2.0 V (Fig. 3.1), revealing stable capacitance up to ~1.8 V, supporting the conclusion that ultraconfined water shifts its

electrochemical stability window beyond the standard 1.23 V. This novel physics phenomenon warrants further dedicated investigations.

Figure 3.1. Capacitance recalculated from cycling voltammograms recorded at 0.025 V/s scan rate for voltage windows ranging from 0–1 V and 0–2 V. Each data point represents 10 repeated cycles; error bars indicate capacitance variation.

6. While graphene is more sustainable than some traditional battery materials, it is not completely eco-neutral—its impact varies based on how it is produced, used, and disposed of. This should be softened in the manuscript.

Our reply: We acknowledge that graphene’s environmental footprint depends heavily on production, use, and disposal methods. The manuscript language has been carefully softened to reflect graphene’s sustainability with appropriate citations [17]. We clarify graphene’s role as a structural component enabling 2D channel uniformity and mention potential replacement with other sustainable carbon materials in future work.

This manuscript presents an exciting and impactful study with strong potential implications for sustainable energy storage. I suggest a minor revision before the acceptance.

Conclusion: We thank the reviewer for recognizing our manuscript’s scientific impact and potential implications for sustainable energy storage. Their suggestions have greatly helped refine our presentation and strengthen our position within the field. We have addressed all comments thoroughly in the revised manuscript and supplementary files.

Summary of Revisions Based on Reviewer 3's Comments

In addition to the changes made in response to Reviewers 1 and 2, the following revisions have been implemented:

1. The Introduction, Discussion, and Conclusion sections have been revised to better benchmark the device.
2. New data on cell instability at high voltages have been added to the Supporting Information (SI).

All-water supercapacitor enabled by 1-nm clay channels

Vasily Artemov*, Svetlana Babiy, Yunfei Teng, Jiaming Ma, Alexander Ryzhov, Tzu-Heng Chen, Lucie Navratilova, Victor Boureau, Pascal Schouwink, Mariia Liseanskaia, Patrick Huber, Fikile Brushett, Lyesse Laloui, Giulia Tagliabue, Aleksandra Radenovic

Point-by-point reply to Reviewers' comments

Reviewer #1 (Remarks to the Author):

I appreciate the authors' thorough response to the initial review and the substantial experimental data and analyses provided in the revised manuscript. While these additions help clarify some aspects of the study, the revised manuscript still does not fully resolve the main scientific concerns raised in the first round of review, particularly regarding the quantitative distinction between the contributions of permanent lattice charge and nanoconfined water to the observed electrochemical behavior. To further improve the scientific rigor and mechanistic clarity, I provide the following suggestions for refinement:

Our reply: We sincerely thank the Reviewer for the careful evaluation of our revised manuscript and for the constructive suggestions aimed at strengthening the mechanistic interpretation. We appreciate the recognition of the additional data provided in the previous round and fully agree that a quantitative distinction between permanent lattice charge and nanoconfined water contributions is central to the scientific rigor of the work.

In this response, we have conducted a new series of comparative, control, in-situ, and temperature-dependent experiments. These collectively demonstrate that (i) capacitance does not scale with lattice charge density, (ii) proton activity is essential for charge storage, and (iii) the electrochemical response is characteristic of ion-type EDL formation in nanoconfined water rather than lattice-charge-driven behavior. All new findings have been incorporated into the revised manuscript (list of changes below).

Below, we address each comment in detail.

Reviewer comment 1: *While the wet/dry MEU comparison and conductivity enhancement under nanoconfinement are convincing, direct quantitative evidence distinguishing the contributions from lattice-charge EDL versus nanoconfined water EDL is still lacking. The following experiments is suggested. Measure clay surface charge density, ζ -potential, or cation exchange capacity (CEC), and compare with capacitance data to estimate the potential contribution of lattice charge. If feasible, perform control experiments using clays without permanent structural charge (e.g., pyrophyllite) to see whether high capacitance persists.*

Our reply: We thank the Reviewer for this important suggestion. We agree that a direct comparison between surface charge metrics and electrochemical capacitance is essential. We have therefore performed a systematic comparative study using clay minerals selected to decouple surface charge from specific surface area (SSA).

A direct comparison between different clays can be misleading if swelling capacity or surface area varies significantly. To avoid this, we selected minerals with comparable SSA but substantially different permanent lattice charge: illite, kaolinite, and talc. Notably, talc

(structurally analogous to the pyrophyllite suggested by the Reviewer) possesses extremely low permanent structural charge while maintaining SSA comparable to kaolinite and illite. Bentonite and montmorillonite were used as high-charge references.

All membrane–electrode units (MEUs) were fabricated using an identical protocol and identical graphene loading. ζ -potentials were measured using colloidal suspensions (Malvern instrument), and CEC values are consistent with literature benchmarks.

Table 1. Surface characteristics of various clays: cation-exchange capacity (CEC) and ζ -potential, compared to the measured MEU capacitance made of these clays.

Clay type →	Bentonite	MMT	Kaolinite	Illite	Talc
CEC	High	High	Medium	Medium	Low
ζ -potential (mV) at neutral pH	-37	-31	-17	-15.7	-20 ¹
Specific surface area (m ² /g) ²	840	800	30	30	20
MEU specific capacitance (F/g _G)	37±2	29±2	8±2	11±6	15±3

Figure 1. Specific capacitance of MEUs at different scan rates for five clay types: bentonite (blue), montmorillonite (red), kaolinite (green), illite (pink), and talc (brown). Specific capacitance of the MEUs was estimated from the voltammograms.

Table 1 quantitatively compares the cation exchange capacity (CEC) and measured specific capacitance, also given in Fig. 1, at different scan rates. The key result is that capacitance does not scale with CEC or ζ -potential. Specifically, talc (very low lattice charge) exhibits

¹ This comparatively high ζ -potential can be understood via re-distribution of the charge between the planes and edges of the clay flakes as discussed, e.g. in Ref. [31] of the manuscript.

² Maček, M. et al., A comparison of methods used to characterize the soil specific surface area of clays, Applied Clay Science, 83–84, 144 (2013).

capacitance comparable to kaolinite and illite. Montmorillonite and bentonite (high CEC) do not show proportionally higher capacitance. Two capacitance groups are observed that correlate with structural confinement, not with surface charge density. If lattice charge were the dominant contributor to EDL formation, capacitance would scale approximately linearly with CEC. This is not observed.

We therefore conclude quantitatively that **permanent lattice charge is not the dominant factor governing the measured capacitance**. Instead, capacitance persists even in low-charge systems, strongly supporting a confinement-controlled EDL mechanism.

Reviewer comment 2: *The current data suggest that H_3O^+/OH^- ions generated in nanoconfined water form the EDL, but additional direct experiments could strengthen the causal link. Use buffered systems that suppress H_3O^+/OH^- generation and monitor the resulting capacitance changes. Employ in-situ characterization techniques (e.g., EQCM or in-situ spectroscopy) to directly track proton migration and accumulation, providing more direct evidence of an ion-type EDL.*

Our reply: We fully agree that additional direct evidence linking proton activity to capacitance is critical. We performed three additional experimental tests: buffer experiments, in-situ electrical impedance spectroscopy (EIS), and temperature-dependent activation energy measurement. All three independently converge on the same conclusion.

- **Buffer-induced suppression of capacitance.** We introduced a 1 M sodium acetate buffer (pH = 7.00 ± 0.05 at 25 °C) into montmorillonite- and talc-based MEUs and compared the result to the same MEUs measured without buffer (Fig. 2). Key observations are: without a buffer (left panels), there is pronounced capacitive CV behavior; with buffer (right panels) capacitance is dramatically suppressed.

Because the buffer regulates proton activity and suppresses free H_3O^+/OH^- fluctuations, this result directly demonstrates that capacitance depends critically on proton availability. If lattice charge were dominant, buffering should not strongly suppress capacitance. The dramatic reduction, therefore, supports the protonic EDL hypothesis.

Figure 2. Cyclic voltammograms (CVs) without (left) and with the buffer (right) for montmorillonite (A) and talc-based (B) MEU.

- **In-situ electrical impedance spectroscopy (EIS).** Measured Nyquist plots for the MMT-based MEU impedance exhibit nearly vertical-going low-frequency ‘tails’ (Fig. 3a). Comparison with the typical curves expected in electrochemical systems (Fig. 4) shows the absence of semicircles associated with Faradaic reactions in our system. This is characteristic of ideal EDL capacitive behavior and not of redox-based charge storage.

Figure 3. (A) Nyquist plots of in-situ electrical-impedance spectroscopy (EIS) for montmorillonite-based MEU measured at different bias voltages (see legend). (B) $Z''(1/\omega)$ graph at low frequencies and capacitances, extracted from slope angles.

[Figure Redacted]

Additionally, from the Z'' vs. $1/\omega$ plot at low frequency (Fig.3b) and modeling the cell as R + C in series, we extract capacitance: 49.6 F g⁻¹ (0 V bias), 28.1 F g⁻¹ (1.0 V), 18.4 F g⁻¹ (1.5 V). These values coincide with those obtained from cyclic voltammetry, confirming internal consistency and ruling out hidden Faradaic contributions, thus explaining the long-term cycling stability of our cells (see MS Fig. 5d).

³ Mei, B.-A. et al., *J. Phys. Chem. C*, 122, 24499 (2018).

⁴ Mei, B.-A. et al., *J. Phys. Chem. C*, 122, 194 (2018).

- Temperature-dependent conductivity and activation energy.** Using a liquid-based temperature controller, we measured impedance of MMT-based MEU as a function of temperature (Fig.5a). Arrhenius analysis of DC conductivity (Fig.5b) yields activation energy = 0.17 ± 0.02 eV. This value is characteristic of proton transport in water and significantly lower than typical migration energies of lattice-bound cations (0.2–0.4 eV). Therefore, the activation energy favors protonic conduction over lattice-controlled ion movement.

Figure 5. (A) Nyquist plots of the MEU measured at different temperatures. The vertical sticks show DC resistivity values Z_{dc} . (B) Arrhenius plot of DC conductivity ($\sigma = 1/Z_{dc}$). The number near the curve is the activation energy.

Summarizing, three independent techniques (buffer suppression, EIS, activation energy) converge to show: charge carriers are protonic (H_3O^+/OH^-), confinement enhances their activity, capacitance originates from ion-type EDL formation in nanoconfined water, and no significant Faradaic or lattice-charge-dominated mechanism is observed.

Reviewer comment 3: *The manuscript explores variations in graphene content and separator thickness, showing effects on capacitance and efficiency. Provide statistical descriptions of interfacial continuity and absence of voids to support reproducibility and the observed high cycling stability.*

Our reply: We thank the Reviewer for emphasizing reproducibility and structural continuity. We performed two additional tests: quantitative porosity analysis and hydration-dependent transparency analysis. Both experiments support manufacturing reproducibility and the lack of interfacial discontinuities.

- Quantitative porosity analysis.** Cross-sectional SEM images were analyzed using ImageJ⁵ for several independently fabricated MEUs with varying graphene content and separator thickness (two examples are in Fig. 6). Binary contrast analysis yielded a narrow pore/area ratio of $\sim 20 \pm 7.5\%$ between the samples. Importantly, the capacitance of the cells varies by orders of magnitude across compositions, while porosity variation remains within $\sim 7.5\%$. Thus, no statistical correlation between pore fraction and capacitance is found. The observed electrochemical trends cannot be attributed to void fraction variations.

⁵ <https://imagej.net/software/fijidistributionsshrinking/>

Figure 6. (A) Example of SEM images of two independently fabricated MEU cross-sections with different concentrations of graphene and different thicknesses of separators. The yellow frame shows a zone processed with ImageJ/Fiji software. (B) Pore size distributions corresponding to the images.

- Hydration-induced transparency change.** The SEM imaging analyzed above is performed under high vacuum, which induces dehydration and possible artificial pore formation in clay structures. To verify this, we compared the optical transparency of dry and hydrated mmt films (Fig. 7). Dry films are opaque (light scattering by voids around a few hundred nm), while hydrated films are transparent. The opacity vanishes upon hydration, demonstrating that large voids collapse when water is present. Since electrochemical measurements are conducted under hydrated conditions, the relevant operational structure is even more compact and continuous than SEM images demonstrate.

Figure 7. (A) Photos of wet and dry bentonite clay films. (B) Pore size distribution at the cross-section of the same films determined from SEM images (not shown).

In conclusion, interfacial continuity is statistically consistent across samples. Porosity variations are too small to explain capacitance differences. Hydrated structures eliminate vacuum-induced voids. High cycling stability further confirms structural integrity.

Reviewer #2 (Remarks to the Author):

The paper has been very strongly improved since the first round of review. Most reviewer's comments have been addressed and many new experiments have been added to support the result. I now think that the paper can be considered for Nature Communications but that important points still have to be addressed.

Overall, the paper's quality does not reflect the quality of the experimental work added in the rebuttal letter. The discussion section is poorly written, shows references to the 'ionic model of water' but without any proper discussion or insight and continuously mixes up transport and dielectric properties.

On top of this, the ion-substitutions in the clay is put forward as key to the mechanism while so much effort have been put into showing that only H⁺ /HO⁻ matter... The authors should focus on what has been well proven experimentally and remove speculations.

Our reply: We sincerely thank the Reviewer for the positive reassessment of our manuscript and for recognizing the substantial experimental improvements introduced in the previous revision. We particularly appreciate the critical remarks regarding the discussion section. We agree that the previous version did not adequately reflect the strength of the experimental evidence and that conceptual distinctions between transport and dielectric properties were not presented with sufficient clarity.

Following the Reviewer's recommendation, we have:

- Completely rewritten the Discussion section.
- Removed speculative interpretations not directly supported by experiments.
- Eliminated ambiguous references to the "ionic model of water" unless experimentally justified.
- Clearly separated proton transport phenomena from dielectric polarization effects.
- Reduced emphasis on ion substitution in the clay lattice, as our experiments demonstrate that protonic species dominate the charge storage mechanism.

The revised Discussion now focuses strictly on experimentally supported conclusions.

Below, we address each comment in detail.

Reviewer comment 1: *The transport mechanism sketched on Figure 2d does not seem compatible with the surface-scaling with exponent 2/3 given in the SI. Or maybe it should be justified.*

Our reply: We thank the Reviewer for raising this important fundamental issue. The apparent inconsistency arises because the two scaling behaviors correspond to different physical regimes. The exponent 2/3 reported in the SI refers to colloidal suspensions below or near the percolation threshold, whereas Figure 2d represents the compacted, percolated regime of the membrane–electrode unit (MEU).

More specifically, electrical conductivity σ in nanoparticle systems follows concentration-dependent scaling: $\sigma \sim c^\alpha$, with α determined by the percolation regime⁶: (1) In the pre-percolation / near-percolation regime, $\alpha < 1$ (including $\approx 2/3$), the conductivity is limited by aggregation, Brownian motion, and electrostatic interactions, and transport pathways are incomplete. (2) Above percolation threshold, $\alpha \approx 1.5\text{--}2.0$, a continuous conductive network is formed, and the exponent depends on particle geometry (elongated particles lead to smaller

⁶ McLachlan, D., Physica B: Condensed Matter, 606, 412658 (2021)

α). (3) In the saturation regime (Figure 2d case), conductive pathways are already established, and σ becomes weakly dependent on concentration.

The SI data correspond to regime (1), while Figure 2d reflects regime (3). These regimes are physically distinct and not expected to obey the same scaling exponent.

We have now clarified this distinction explicitly in the revised manuscript and added a short explanatory paragraph in the SI to avoid confusion. We agree that a full percolation analysis would be interesting, but would constitute a separate study beyond the scope of the present work.

Reviewer comment 2: *The ‘competitive parameters’ of the device are not justified. Supercapacitors are a well-established field. Authors should benchmark more properly their device with respect to this literature.*

Our reply: We thank the Reviewer for this important point. We agree that the term 'competitive' was not appropriately justified in the previous version and could be misleading. We have therefore removed the term 'competitive' throughout the manuscript and replaced it with a structured benchmarking section. Our system differs conceptually from conventional supercapacitors in two fundamental ways: (1) water acts as the sole electrolyte; (2) charge storage arises from nanostructure-induced interfacial proton activity, not bulk electrolyte optimization.

We now position the device as a proof-of-concept architecture demonstrating confinement-enabled protonic EDL formation in water-only systems. Rather than claiming superiority in absolute performance, we highlight eco-neutral material composition, absence of flammable organic electrolytes, structural simplicity, and scalability potential. This reframing aligns the manuscript with realistic benchmarking.

Reviewer comment 3: *(a) Confinement usually refers to the geometric constraint in contrast to surface effects which denote the liquid’s behavior at a single surface. Here the observed effect is due to surface effects and that narrow channels merely allow to increase density by removing bulk liquid regions. Thus, I think it should not be called ‘confinement’ but interfacial or surface effects. See discussion in Ref⁷.*

Our reply: We appreciate this conceptual distinction highlight. In the revised manuscript, we now define confinement explicitly as: A condition in which water properties become spatially heterogeneous due to geometric restriction, leading to symmetry breaking and anisotropy in electrical properties. This definition encompasses steric (geometric) restriction, surface-induced ordering, and interfacial polarization. We emphasize that in real nanoporous systems, steric and surface effects are intrinsically entangled and cannot be experimentally separated in our architecture. Importantly, as shown in our response to Reviewer #1, capacitance does not correlate with lattice charge density, indicating that the effect is not purely surface-charge driven.

We have clarified terminology throughout and cited the reference suggested by the Reviewer.

⁷ Wang, Y. et al., Nat. Commun. 16, 7288 (2025).

(b) - *This comment actually leads to one of main claims of the paper which I think is wrong: that any type of confinement will split water and increase capacitance. Although the authors show several examples in the rebuttal letter, I think this clearly would not happen with clean graphite surfaces which are not able to split water just to accommodate the surface charge. Here it seems that the authors are using acidic surfaces which tend to improve water splitting (or proton activity as they write).*

Our reply: We agree with the Reviewer that our previous wording overstated universality. We have removed any statements suggesting that *any* confinement necessarily induces water splitting. Instead, we now state:

Under the specific physicochemical conditions studied here—hydrated clay nanochannels—confinement correlates with enhanced protonic conductivity and EDL capacitance.

We also acknowledge that interfaces such as pristine graphite may behave differently depending on surface chemistry and electronic structure. We have softened the universality claim throughout the manuscript and limited conclusions strictly to experimentally tested systems. Regarding recent reports of anomalous proton conduction at graphite interfaces⁸, we now cite the relevant work and frame it as supportive but not definitive evidence of broader applicability.

(c) - *Overall I think that emphasizing ‘confinement’ is wrong and detrimental to the paper. See results on graphite confinement⁹.*

Our reply: We respectfully maintain that confinement, as defined above, remains the most appropriate term to describe isolation of interfacial water layers, suppression of bulk-like symmetry, and enhanced protonic transport. However, we have removed overly strong claims, clarified definitions, and acknowledged alternative interpretations (surface-dominated models). The revised manuscript presents confinement as a structural condition rather than a universal causal law.

Reviewer comment 4: *The comment linking the high water-splitting overpotential to confinement vs bulk effects is absurd. It merely shows weak catalytic activity of these materials for OER and HER, there is nothing to do with confined water here.*

Our reply: We agree. The statement linking voltage window extension to confinement has been completely removed. We now simply report the extended voltage window as an experimental observation without a mechanistic attribution. Speculative discussion has been eliminated.

Reviewer comment 5: *Cycling units in days on Figure 5d ? This is not serious, use ‘number of cycles’.*

Our reply: We fully agree. The figure has been corrected to display the number of cycles instead of days.

Reviewer comment 6: *Throughout the paper, the authors claim to use graphene for the electrodes, graphene pristine enough to be hydrophobic. I guess this is some graphene oxide (GO) solution that is*

⁸ Wang, R. et al. In-plane dielectric constant and conductivity of confined water, *Nature* 646, 606 (2025).

⁹ Esfandiari, A. et al. Size effect in ion transport through angstrom-scale slits. *Science* 358, 511–513 (2017).

somehow reduced. Could the authors show Raman spectra to establish whether this is graphene or GO?

Our reply: We thank the Reviewer for this important request. We performed additional Raman spectroscopy measurements comparing: graphite reference, the graphene dispersion used in electrode fabrication, and two graphene–clay composites with different graphene loadings. The spectra (Fig. 8) show: prominent D and G bands and an increased D/G ratio relative to graphite. These features are consistent with reduced graphene oxide (rGO) rather than pristine graphene. We have corrected the manuscript accordingly and now refer to the electrode material as reduced graphene oxide (rGO). The Raman spectra are included in the SI (new Figure S31).

Figure 8. Raman spectra of graphite (black line), graphene solution used for electrodes (green line), and full MEU assemblies with different graphene concentrations (blue and red lines).

Reviewer comment 7: *Ref 27 is wrong*

Our reply: We thank the Reviewer for identifying the error. The incorrect title has been corrected. We apologize for this oversight.

Reviewer comment 8: *Comparison with photosynthesis seems irrelevant, or at least should be justified*

Our reply: We appreciate this comment. Our intention was not to imply functional similarity with photosynthesis but to illustrate a design philosophy that biological systems prioritize material abundance and functional sufficiency, while industrial energy systems often optimize peak performance using rare materials. To avoid misunderstanding, we have clarified the analogy and shortened its discussion. Explicitly stated that the comparison refers only to materials design strategy, not mechanistic similarity.

Overall, we believe the revised manuscript now reflects the quality and rigor of the experimental work and aligns fully with the Reviewer's expectations.

We thank the Reviewer for the insightful and constructive feedback, which has significantly strengthened the clarity and scientific robustness of the manuscript.

Reviewer #3 (Remarks to the Author):

I was the reviewer of the initial submission of this manuscript and originally recommended minor revisions. The authors have substantially improved the paper, comprehensively addressing all previously raised points and significantly strengthening the presentation, clarity, and contextualization of their results.

The study presents a highly novel concept: an all-water supercapacitor based on ultraconfined 1-nm water channels formed within naturally abundant clays. The work elegantly demonstrates that confined water can act as a sole electrolyte in a scalable macroscopic device, achieving performance metrics (specific capacitance, coulombic efficiency, and exceptional cycling stability) that are highly competitive with traditional systems. Importantly, these results are achieved using inexpensive, environmentally benign, and abundant materials such as montmorillonite clay, demonstrating clear potential for sustainable, large-scale applications.

While research relying on naturally abundant materials is often challenging due to issues related to purity, heterogeneity, and secondary effects, the authors convincingly overcome these hurdles. The manuscript provides thorough physico-chemical characterization, rigorous cleaning protocols, and multiple independent validation techniques. Furthermore, the work is well supported by a strong theoretical framework and by the authors' studies on clean, idealized systems (e.g., nanoconfined water between diamond/graphene channels). Compared with the initial submission, the revised manuscript is clearer, better substantiated, and more impactful.

Given the substantial improvements and the high scientific and technological originality of the work, I recommend acceptance as is.

Our reply: We are sincerely grateful to the Reviewer for the careful evaluation of both the original and revised versions of our manuscript, and for the continued support throughout the review process. We particularly appreciate the recognition of the conceptual novelty of the all-water supercapacitor architecture and of the effort invested in strengthening the experimental validation, clarity, and contextualization of the results. It is especially encouraging that the Reviewer highlights the successful translation of insights obtained in idealized systems—such as nanoconfined water in well-defined diamond/graphene channels—to a scalable platform based on naturally abundant materials such as clay.

We are also thankful for the acknowledgment of the challenges associated with using naturally sourced materials, including heterogeneity and impurity control. Considerable effort was devoted to purification protocols, structural and physicochemical characterization, and independent validation methods to ensure reproducibility and mechanistic clarity. We are pleased that these efforts are recognized as convincing and rigorous.

Summary of revisions based on reviewers' #1 and #2 comments

[All the changes mentioned below are marked with red in the modified manuscript]

Abstract

- Replaced precise voltage wording with “~1.6 V” to avoid overstating precision.
- Rephrased cycling stability to “*stable performance over more than 60,000 cycles.*”
- Added concluding statement emphasizing the main result.

Introduction

- Clarified the statement about atmospheric electrification.
- Reframed the motivation toward sustainable energy storage strategies.
- Clarified the novelty of the concept as using pure water for reversible charge storage.
- Emphasized environmentally benign materials as a motivation for the study.
- Moderated claims about nanoconfinement effects by replacing definitive language with “*can deviate from bulk behavior.*”
- Removed a strong claim about challenging classical electrodynamics.
- Added concluding statement highlighting geometric confinement rather than chemical complexity as the design principle.

Results

- Clarified that hydrated clays form extended networks of nanometer-scale aqueous channels supporting proton transport.
- Replaced wording implying “*competitive parameters*” with “*stable electrochemical performance.*”
- Removed speculative lifetime extrapolation (“*150 years*”) and replaced with explanation that long cycle stability.
- Figure 2d corrected accordingly.
- Rephrased electrolysis threshold statement to a neutral observation of stable operation up to ~1.6–1.65 V rather than attributing it to confinement.
- Removed comparison with photosynthesis efficiency.

Discussion

- Rewritten the Discussion section entirely.
- Rewrote the opening paragraph to clearly attribute electrochemical behaviour to water confined in nanometre-scale clay channels.
- Introduced a conservative interpretation of the mechanism based on electrical double-layer (EDL) formation.
- Added discussion explaining how nanoconfinement modifies water structure and transport properties.
- Removed speculative mechanistic claims.
- Clarified that the microscopic transport pathway is not yet resolved.
- Separated transport from dielectric effects.
- Added experimental evidence description supporting the mechanism:
 - o electrochemical impedance spectroscopy,
 - o suppression of capacitance by pH buffer,
 - o temperature-dependent activation energy (~0.17 eV).
- Clarified that protonic species (H_3O^+ and OH^-) are the likely charge carriers.
- Reduced emphasis on ion substitution in clays.
- Clarified that structural lattice charge in clays is not the dominant factor controlling the electrochemical response.
- Softened universality statements.

- Improved benchmarking.
- Clarified how the system differs from conventional supercapacitors using bulk electrolytes.
- Framed the device as a proof-of-concept platform for sustainable electrochemical systems.
- Removed unsupported claims regarding overpotential.
- Added a concluding statement noting that a complete microscopic description of transport requires further study.

Conclusion

- Added a statement emphasizing that nanoconfined water in layered materials can support macroscopic electrochemical functionality.
- Framed the work as a proof-of-concept platform for future sustainable aqueous electrochemical systems.

Methods

- Added talc to the list of tested clay minerals.
- Clarified that purified clay suspensions and vapor-phase hydration minimize dissolved salts, supporting the interpretation that the response arises from confined water.

Acknowledgements

- Added acknowledgement of Claudia Goy, Alexander Petrov, and Samira Afsharian for assistance with Raman and optical measurements.

References

- Removed old references [19-23], and [40].
- Added new references [34], [35], and [39].

Supporting Information

- New Figures S31 to S36 added with the captions.
- The front matter adjusted accordingly.

All-water supercapacitor enabled by 1-nm clay channels

Vasily Artemov*, Svetlana Babiy, Yunfei Teng, Jiaming Ma, Alexander Ryzhov, Tzu-Heng Chen, Lucie Navratilova, Victor Boureau, Pascal Schouwink, Mariia Liseanskaia, Patrick Huber, Fikile Brushett, Lyesse Laloui, Giulia Tagliabue, Aleksandra Radenovic

Point-by-point reply to Reviewers' comments

Reviewer #1 (Remarks to the Author):

The authors have adequately addressed my concerns, and the revised manuscript is now scientifically sound. I recommend acceptance of the manuscript in its current form.

Our reply: We thank the reviewer for the positive evaluation of the revised manuscript and for recommending acceptance.

Reviewer #2 (Remarks to the Author):

The manuscript has improved again since the last round of review. I acknowledge the effort put to answer to reviewers. Still, the authors have to be more rigorous in the discussion section. The activation energy of 0.17eV is that of water self-diffusion. Thus indeed, one can argue that the transport mechanism is not hopping at surfaces and instead a diffusion process in liquid. However, one cannot claim that it's due to structural diffusion (what authors call relay race) instead of conventional vehicular diffusion, since both have 0.17eV for an activation energy. The authors have to be careful because throughout the review process, their overclaims and lack of justifications have constantly undermined my trust in their work.

Besides this comment, I now think the paper is good enough for publication.

Our reply: We thank the reviewer for the careful assessment of the revised manuscript and for the constructive comments on the interpretation of the activation energy.

In response, we have revised the Discussion to provide a more conservative interpretation of the measured activation energy (0.17 eV). We now state that this value is consistent with proton-related transport in liquid water and is also in agreement with reported values for water self-diffusion. We further clarify that this activation energy alone does not allow discrimination between structural diffusion (the “relay-race” mechanism) and conventional vehicular diffusion, as both processes may exhibit similar energetic barriers.

Accordingly, statements implying a specific microscopic mechanism have been removed or rephrased to avoid overinterpretation. The discussion has been revised to ensure that all conclusions remain strictly supported by the experimental evidence. A relevant supporting reference [37] has been added.